# Novel Alzheimer risk genes determine the microglia response to amyloid-β but not to TAU pathology

Annerieke Sierksma[1,2,†] ID, Ashley Lu[1,2,†] ID, Renzo Mancuso[1,2] ID, Nicola Fattorelli[1,2] ID, Nicola Thrupp[1,2], Evgenia Salta[1,2], Jesus Zoco[1,2] ID, David Blum[3] ID, Luc Buée[3], Bart De Strooper[1,2,4,*,‡] ID & Mark Fiers[1,2,‡,**] ID

## Abstract

Polygenic risk scores have identified that genetic variants without genome-wide significance still add to the genetic risk of developing Alzheimer's disease (AD). Whether and how subthreshold risk loci translate into relevant disease pathways is unknown. We investigate here the involvement of AD risk variants in the transcriptional responses of two mouse models: APPswe/PS1[L166P] and Thy-TAU22. A unique gene expression module, highly enriched for AD risk genes, is specifically responsive to Aβ but not TAU pathology. We identify in this module 7 established AD risk genes (*APOE, CLU, INPP5D, CD33, PLCG2, SPI1,* and *FCER1G*) and 11 AD GWAS genes below the genome-wide significance threshold (*GPC2, TREML2, SYK, GRN, SLC2A5, SAMSN1, PYDC1, HEXB, RRBP1, LYN,* and BLNK), that become significantly upregulated when exposed to Aβ. Single microglia sequencing confirms that Aβ, not TAU, pathology induces marked transcriptional changes in microglia, including increased proportions of activated microglia. We conclude that genetic risk of AD functionally translates into different microglia pathway responses to Aβ pathology, placing AD genetic risk downstream of the amyloid pathway but upstream of TAU pathology.

**Keywords** Alzheimer's disease; genetic risk; microglia; RNA-seq; single cell
**Subject Categories** Chromatin, Transcription & Genomics; Neuroscience

## Introduction

Genetic background strongly determines the risk of sporadic Alzheimer's disease (AD) (Gatz *et al*, 2006). Unlike the APOE4 polymorphism and 42 other genetic loci, thousands of SNPs associated with risk of AD do not reach genome-wide significance (Efthymiou &

Goate, 2017; Marioni *et al*, 2018; Verheijen & Sleegers, 2018). Polygenic risk scores (PRSs) incorporate the contributions of these variations and relate these contributions to disease risk (Purcell *et al*, 2009). PRSs for AD currently reach a prediction accuracy of 84%, albeit a major proportion can be attributed to APOE status alone (Escott-Price *et al*, 2017). Two crucial questions arise from the myriad of genetic studies: (i) Are AD risk genes functionally linked to amyloid-β (Aβ) or to TAU pathology and (ii) do they converge within a single cellular or molecular pathway, or do they define many parallel pathways that all lead to AD? Although it remains challenging to causally link AD risk loci to affected genes, we expect that at least part of the genes and pathways implicated in genome-wide association studies (GWAS) affect the cellular response of the brain to Aβ or TAU pathology (De Strooper & Karran, 2016; Efthymiou & Goate, 2017). Such a model integrates parts of the amyloid hypothesis with the complex genetics of AD, which will lead to a more coherent view on the pathogenesis of AD.

Profiling of postmortem brain tissue only provides insights into the advanced stages of AD and cannot delineate cause–consequence relationships, which is required to develop mechanistic models for the pathogenesis of AD (Zhang *et al*, 2013). Transgenic mouse models on the other hand only partially recapitulate AD or frontotemporal dementia (FTD) phenotypes, but they provide detailed functional insights into the initial steps of disease, which is of high relevance for preventative therapeutic interventions (Zahs & Ashe, 2010). What is lacking until now, however, is the integration of functional information from mouse studies with the GWAS data obtained in human. Doing so may help to determine whether subsignificant AD risk loci are involved in the cellular responses to Aβ or TAU pathology. This would increase confidence that these genes are truly involved in AD and indicate within which pathways these genes play a functional role.

Here, we perform transcriptional profiling of mouse hippocampus after exposure to Aβ or TAU pathology, at early (4 months of age; 4M) and mature stages of disease (10-11M). We use APP[swe]/

1  VIB Center for Brain & Disease Research, Leuven, Belgium
2  Laboratory for the Research of Neurodegenerative Diseases, Department of Neurosciences, Leuven Brain Institute (LBI), KU Leuven (University of Leuven), Leuven, Belgium
3  INSERM, CHU Lille, LabEx DISTALZ, UMR-S 1172, Alzheimer & Tauopathies, Université Lille, Lille, France
4  UK Dementia Research Institute, University College London, London, UK
   *Corresponding author. Tel: +32 4957 71044; E-mail: b.strooper@ukdri.ucl.ac.uk
   **Corresponding author. Tel: +32 4944 95150; E-mail: mark.fiers@kuleuven.vib.be
   †These authors contributed equally to this work as first authors
   ‡These authors contributed equally to this work as senior authors

PS1$^{L166P}$ (APPtg) and Thy-TAU22 (TAUtg) mice, both expressing the transgene from a *Thy1.2* promotor (Radde *et al*, 2006; Schindowski *et al*, 2006). The biochemical insults mimicked in these two animals reflect the two main morphological hallmarks of sporadic (SAD) and familial cases of AD (FAD). Therefore, these FAD mouse models are useful models to assess the cellular and transcriptional response to this biochemical pathology. We demonstrate that despite similar robust cognitive phenotypes, APPtg mice develop severe age-dependent transcriptional deregulation, while TAUtg mice have a milder and over time more stable molecular phenotype. AD risk genes uniquely converge in APPtg mice into a coordinated deregulated multicellular gene network that is strongly enriched in neuroinflammatory functions while in the Tau model we mainly observe alterations in genes involved in neuronal biology. Single microglia sequencing confirms this strong and early neuroinflammation in APPtg mice, demonstrating more activated response microglia (ARM; 57%) than homeostatic microglia (20%) in APPtg-11M mice, whereas these phenotypic shifts were much less pronounced in TAUtg-11M mice. Our work provides evidence that a large part of the genetic risk of AD determines the microglial response to Aβ and promotes 11 candidate GWAS variants for future functional AD research.

## Results

At 4M of age, APPtg and TAUtg mice are cognitively intact with mild levels of pathology, whereas they display at 10M overlapping profiles of hippocampus-dependent mnemonic deficits and substantial pathology (see Fig 1A; Radde *et al*, 2006; Schindowski *et al*, 2006; Lo *et al*, 2013). RNA-seq was performed on the hippocampus of 4M and 10M APPtg and TAUtg mice (TG) and their respective wild-type (WT) littermates, with $n = 12$ per group and $n = 96$ in total, yielding on average 7.7 million reads per sample (see Fig 1B). Bulk RNA-seq provides good insights into alteration of functional pathways upon disease onset and progression and allows us to uncover co-regulation beyond individual cell types.

### Enrichment of AD risk genes is only found in the transcriptomic response of APPtg mice

A 2 × 2 linear model (Fig 1C) was employed to investigate the effects of genotype, age, and age*genotype interaction. The age comparison identifies transcripts that change between 4M and 10M mice within each strain (i.e., TG and WT mice of 4M are compared to TG and WT mice of 10M, analyzing the APPtg and TAUtg strains separately; see Fig 2A). The genotype comparison shows mRNA differences between WT and TG mice (Fig 2B). The age*genotype interaction, finally, assesses which transcripts change with aging uniquely in the TG mice (Fig 2C). The study thus reflects the transcriptional changes manifesting in the mice at two critical time points: initially when the first signs of Aβ and TAU pathology occur, and later on, when biochemical alterations manifest with accompanying cognitive deficits.

We wondered whether GWAS-based AD risk variants would be equally responsive to Aβ or TAU pathology. We included established AD risk variants, i.e., loci with $P < 5 \times 10e-8$ in various GWAS studies, as well as subthreshold AD risk variants as these

contribute significantly to AD risk predictions through polygenic inheritance (Escott-Price *et al*, 2015, 2017). We examined multiple sets of such genes taken from Marioni *et al* (2018), which combines UK Biobank AD-by-proxy data with the IGAP database and which confers risk loci onto genes based on proximity (thus from here on, AD risk variants are referred to as AD risk genes, noticing that this is based on these assumptions). Using arbitrary Bonferroni-adjusted *P*-value ($P_{mar}$) cut-offs with decreasing significance for AD association, gene sets of increasing size were created (see Fig 1D and Appendix Table S1). PRS studies have demonstrated that GWAS SNPs up to a cut-off of $P < 0.5$ still improve the predictive power of the risk score (Escott-Price *et al*, 2015, 2017). We decided to limit our study to a cut-off $P_{mar} < 0.05$, yielding an already large set of 1,799 genes. The enrichment of these AD risk gene sets in the different transcriptional responses of the mice was assessed using gene set enrichment analysis (GSEA; Subramanian *et al*, 2005). The data (Fig 1D) demonstrate that independently of the set size, ranging from 92 genes ($P_{mar} < 5e-6$) to 1,799 genes ($P_{mar} < 5e-2$), AD risk genes are found consistently, and significantly ($P_{adj} < 1e-250$) enriched among the genes changing as APPtg mice age ("APPtg A*G" Fig 1D), but not in TAUtg mice. The smallest set ($n = 92$ genes with $P_{mar} < 5e-06$) contains many microglia-expressed genes, e.g., *Treml2*, *Inpp5d*, or *Gal3st4*, (see Fig 2D and Dataset EV1). Thus, genes that enhance the risk of AD are clustering among genes that are deregulated over time with increasing Aβ but not TAU pathology.

### Changes in gene expression exacerbate with aging in APPtg but not in TAUtg mice

To assess the functional effect of the AD risk gene enrichment in APPtg mice, we compared the transcriptional deregulation in the two mouse models in more detail (see Fig 2A–C and Dataset EV1). The transcriptional response of the APPtg and TAUtg mice caused by aging (i.e., independent of transgene) is practically identical (Spearman correlation $R = +0.95$, $P = 1.3e-29$, 95% confidence interval (CI) +0.91 to +0.97; see Fig 2A). When comparing the effects of transgene expression only, the similarity between APPtg and TAUtg mice becomes rather moderate ($R = +0.50$, $P = 1.1e-19$, 95% CI = +0.41 to +0.58; see Fig 2B) and this is only slightly enhanced in the interaction model of age*genotype ($R = +0.67$, $P = 1e-106$, 95% CI = +0.63 to +0.71; see Fig 2C). Thus, while both mouse models age in similar ways, major differences in the transcriptional response between APPtg and TAUtg mice show that these are very different pathologies causing very divergent cellular reactions.

The *APP/PSEN1* transgene causes prominent changes (287 genes in total) in gene expression (green dots, Fig 2B) with most genes ($n = 219$, i.e., 76%) upregulated [log2 fold change (LFC): +0.07 to +5.00, Benjamini–Yekutieli-adjusted *P*-value ($P_{adj}$) < 0.05]. When the aging component is added (i.e., age*genotype), even more genes become upregulated [623 mRNAs (i.e., 78%), LFC: +0.12 to +2.98, $P_{adj} < 0.05$], while 175 genes downregulate their expression (LFC: −0.67 to −0.08, $P_{adj} < 0.05$; Fig 2C). The many upregulated genes in APPtg are often involved in microglia and neuroinflammatory responses, including *Tyrobp* (LFC genotype (G): +1.19, LFC age*genotype (A*G): +1.53), *Cst7* (LFC G: +5.00, LFC A*G: +2.62), and *Itgax* (LFC G: +3.22, LFC A*G: +2.24). These changes are

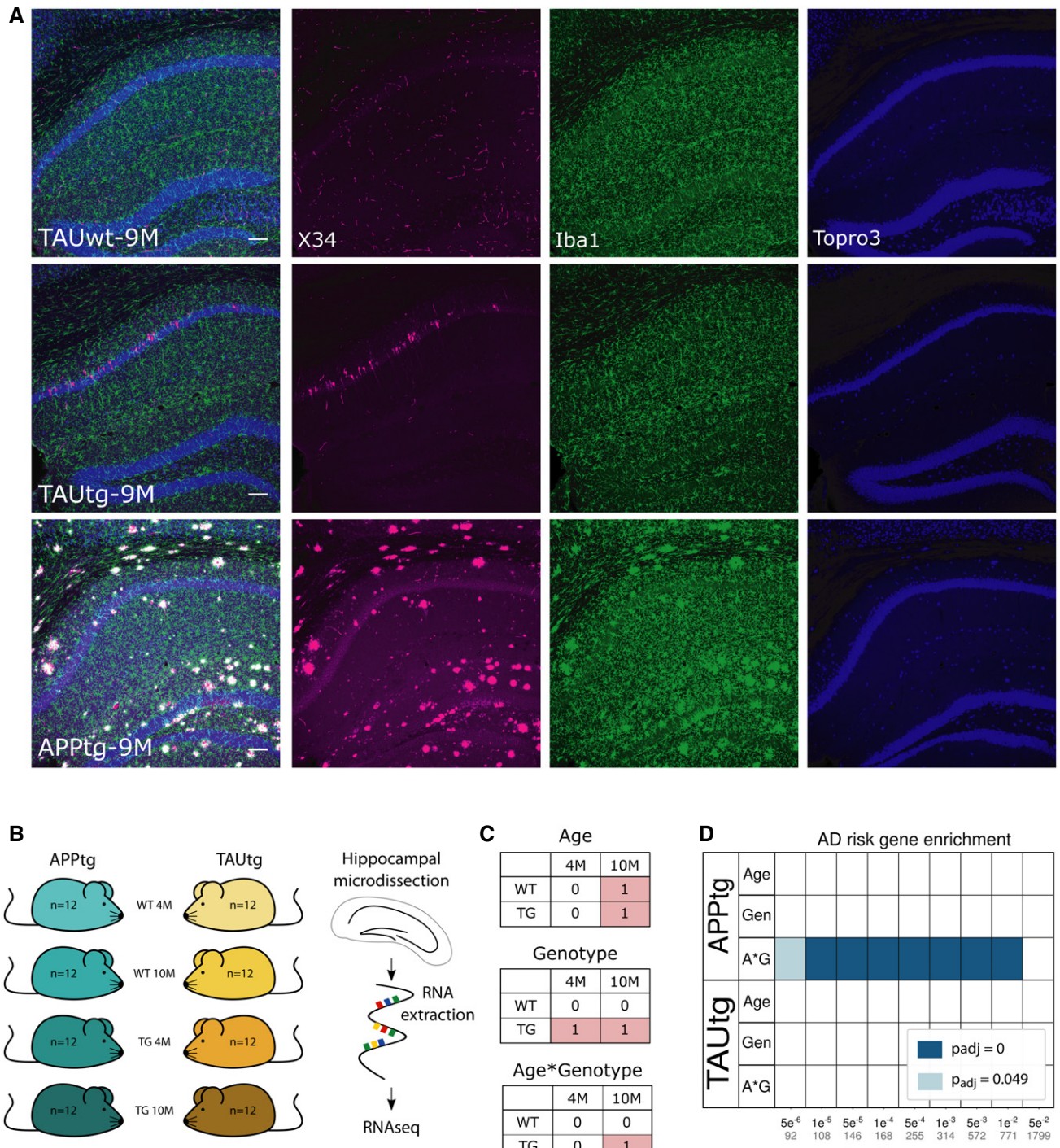

**Figure 1. Enrichment of AD risk genes in APPtg and not in TAUtg mice.**

A  Visualization of the pathology load in TAUwt, TAUtg, and APPtg at 9M of age. Immunofluorescent staining for X34 (a fluorescent derivative of Congo Red; in magenta), Iba1 (microglia; in green), and TO-PRO-3 (nuclei; in blue) has been pseudocolored. Scale bar = 100 μm.

B  Experimental design for mRNA sequencing using $n$ = 12 per experimental group.

C  Explanation of the 2 × 2 linear model, where those cells labeled with 1 are compared to the cells labeled with 0. In the age comparison, mRNA expression in all 10-month-old (10M) mice is compared to all 4-month-old (4M) mice. In the genotype comparison, mRNA expression in all transgenic (TG) mice is compared to all wild-type (WT) mice. In the age*genotype comparison, we assess which transcripts are differentially expressed in the 10M TG mice compared to all other groups.

D  Based on Marioni *et al* (2018), various sets of AD GWAS risk genes were created using different cut-off *P*-values indicated on the *x*-axis (number of genes within each set is written in gray). Enrichment for AD risk genes was assessed among the different statistical comparisons for APPtg and TAUtg mice (A*G: age*genotype, Gen: genotype), using gene set enrichment analysis (GSEA, Subramanian *et al*, 2005). Colors represent the Benjamini–Yekutieli-adjusted *P*-value for the enrichment; blank means no significant enrichment.

strong, up to 32-fold. Indeed, by using gene sets specific for the different brain cell types (Zeisel *et al*, 2015), we can verify that APPtg mice show an increase of up to 80% in microglia-specific transcripts at 10M of age (see Fig EV1). The enriched AD risk genes (Fig 1D) are predominantly upregulated in the age*genotype comparison (Fig 2D). Thus, it appears that an increased expression of neuroinflammatory genes, most likely of microglial source, is an early (Fig 2B) and persistent (Fig 2C) component of Aβ pathology and many AD risk genes follow a similar transcriptional response. As we discuss below, these changes have a complex origin being partially explained by microgliosis in the APPtg mouse (Radde *et al*, 2006), but also a consequence of changes in microglial cell states (Keren-Shaul *et al*, 2017; Krasemann *et al*, 2017; Sala Frigerio *et al*, 2019).

TAUtg mice show markedly fewer transcriptional changes with very little aggravation over time. In the genotype comparison (TAUwt versus TAUtg), only 47 genes become significantly upregulated (LFC: +0.06 to +1.52; $P_{adj} < 0.05$) and 77 downregulated (LFC: −1.30 to −0.05, $P_{adj} < 0.05$; Fig 2B, yellow dots). Only 9 genes are upregulated (LFC: +0.25 to +0.60, $P_{adj} < 0.05$; Fig 2C, red dots) when genotype is combined with age in the interaction model. The majority of deregulated genes (62%) in TAUtg versus TAUwt show decreased expression (Fig 2B) and are of neuronal origin (Fig EV1). The 9 upregulated genes in the age*genotype comparison of TAUtg mice overlap with upregulated genes in APPtg mice and are of microglial (*C1qa, C1qc, Tyrobp, Ctss, Irf8, Mpeg1, Cst7, Rab3il1*) or astroglial (*Gfap*) origin. The overlap with APPtg reflects a (much milder) microglia response in TAUtg. With the exception of *Cst7* (LFC: 2.08), the upregulation is indeed very modest (average LFC of 8 others: 0.38) compared to APPtg mice (max LFC: 2.98; average LFC: 0.70). Similarly, cell type-specific gene expression demonstrates a modest increase in microglial and astrocytic transcripts at older ages, but an early and persistent loss of neuronal and synaptic transcripts in TAUtg mice (see Fig EV1).

Overall, we can conclude that the molecular, pathobiological, and cellular responses in APPtg and TAUtg are fundamentally different despite exhibiting very similar cognitive phenotypes (Radde *et al*, 2006; Schindowski *et al*, 2006; Lo *et al*, 2013). APPtg drives a strong, over time exacerbating inflammatory response, while the TAU transgene causes an early effect on gene expression of genes related to neuronal functions. Most importantly, genes associated with genome-wide statistical significance to AD as well as genes below the significance threshold are transcriptionally active when facing accumulating Aβ, but not TAU pathology.

## AD risk genes are co-regulated in a specific functional gene expression module

Next, we performed unbiased weighted gene co-expression network analysis (WGCNA; Zhang & Horvath, 2005; Langfelder & Horvath, 2008) on each mouse model separately to investigate whether AD risk genes would cluster into functional modules. We obtained in total 63 modules (Appendix Figs S2 and S3). GSEA with the GWAS gene set generated from Marioni *et al* (2018), at different cut-offs for statistical significance as explained above (Fig 3A and Appendix Table S1), demonstrated that the largest set of risk genes (e.g., $n = 1,799$ genes at $P_{mar} < 0.05$) enrich among 4 APPtg- and TAUtg-based modules (Turquoise, Blue, Fig 3A and Appendix Figs S2 and S3). However, when taking gene sets defined by increasing statistical significance ($P_{mar} < 0.001$ or smaller), the only module that persistently demonstrates significant enrichment with GWAS genes is the APPtg-Blue module (see Fig 3A). The APPtg-Blue module is large ($n = 4,236$) and contains 62% of all the genes significantly differentially expressed in aging APPtg mice [age*genotype, which is more than expected by chance (log2 odds ratio (LOR): 2.90, $P = 1.54e$-158)]. We assume that this module provides the integrated and coordinated response of the brain to amyloid pathology. It is important to stress that despite the module being large, still significantly more AD GWAS genes are found in the APPtg-Blue module than expected by chance (1e-250 < $P_{adj} < 0.01$). Thus, it appears that genes associated with increasing risk of AD generally cluster within this module.

Remarkably, most of the differentially expressed genes in this module are upregulated (Fig 3B and C). We functionally characterized this Aβ-induced transcriptional response. The APPtg-Blue module shows a highly significant overlap with the microglia-specific gene set defined by Zeisel *et al* (2015) (LOR: 1.90, $P_{adj} = 1.74e$-77; Fig 4B and Appendix Table S2) and to a lesser extent the astrocyte gene set (LOR: 0.54, $P_{adj}$: 0.014). Moreover, this module is highly enriched for GO categories involving immune response, cytokine production, and inflammation (Fig 3D and Datasets EV2 and EV3). It furthermore shows a highly significant overlap with a recently published Aβ-response network (Salih *et al*, 2019; LOR: 2.0, $P = 2.2e$-16) as well as with the microglia–immune module derived from the brains of late-onset AD patients (Zhang *et al*, 2013; LOR: 2.14, $P = 8.0e$-73), demonstrating that this transcriptional response to increasing Aβ is similar in this Aβ mouse model and AD patients.

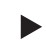

---

**Figure 2. Changes in gene expression exacerbate with aging in APPtg but not in TAUtg mice.**

Log2 fold change (LFC) in TAUtg (*x*-axis) and APPtg mice (*y*-axis) after differential expression analysis. Upregulated genes are on the right part (TAUtg mice) or upper part (APPtg mice) of the graph; downregulated genes are on the left part (TAUtg) or lower part (APPtg) of the graph. Colored dots represent significantly differentially expressed genes (Benjamini–Yekutieli-adjusted *P*-values ($P_{adj}$) < 0.05) for APPtg (green dots), TAUtg (yellow dots), or for both (red dots). Spearman correlation assesses the correlation between APPtg and TAUtg mice when ranking genes that are significantly differentially expressed in either APPtg or TAUtg mice from most up- to most downregulated on a combined score of LFC and $P_{adj}$ (i.e., signed log10(*P*-value)), where the sign is determined by the LFC.

A   Genes differentially expressed as a function of age, i.e., 4M versus 10M, independently of genotype. Thus, genes with a positive LFC are more highly expressed in 10M mice over 4M mice.

B   Genes differentially expressed due to genotype, i.e., WT versus TG, independently of age. Thus, genes with a positive LFC are more highly expressed in TG mice than in WT mice.

C   Genes differentially expressed in the age*genotype interaction comparison, i.e., comparing the TG-10M mice to all other experimental groups (see Fig 1C). Thus, genes with a positive LFC are more highly expressed in the TG-10M mice compared to all other experimental groups.

D   Depicts the 314 Marioni-based GWAS genes at $P < 0.001$ onto the LFC/LFC plot of the age*genotype comparison (panel C). Green dots are significantly changed.

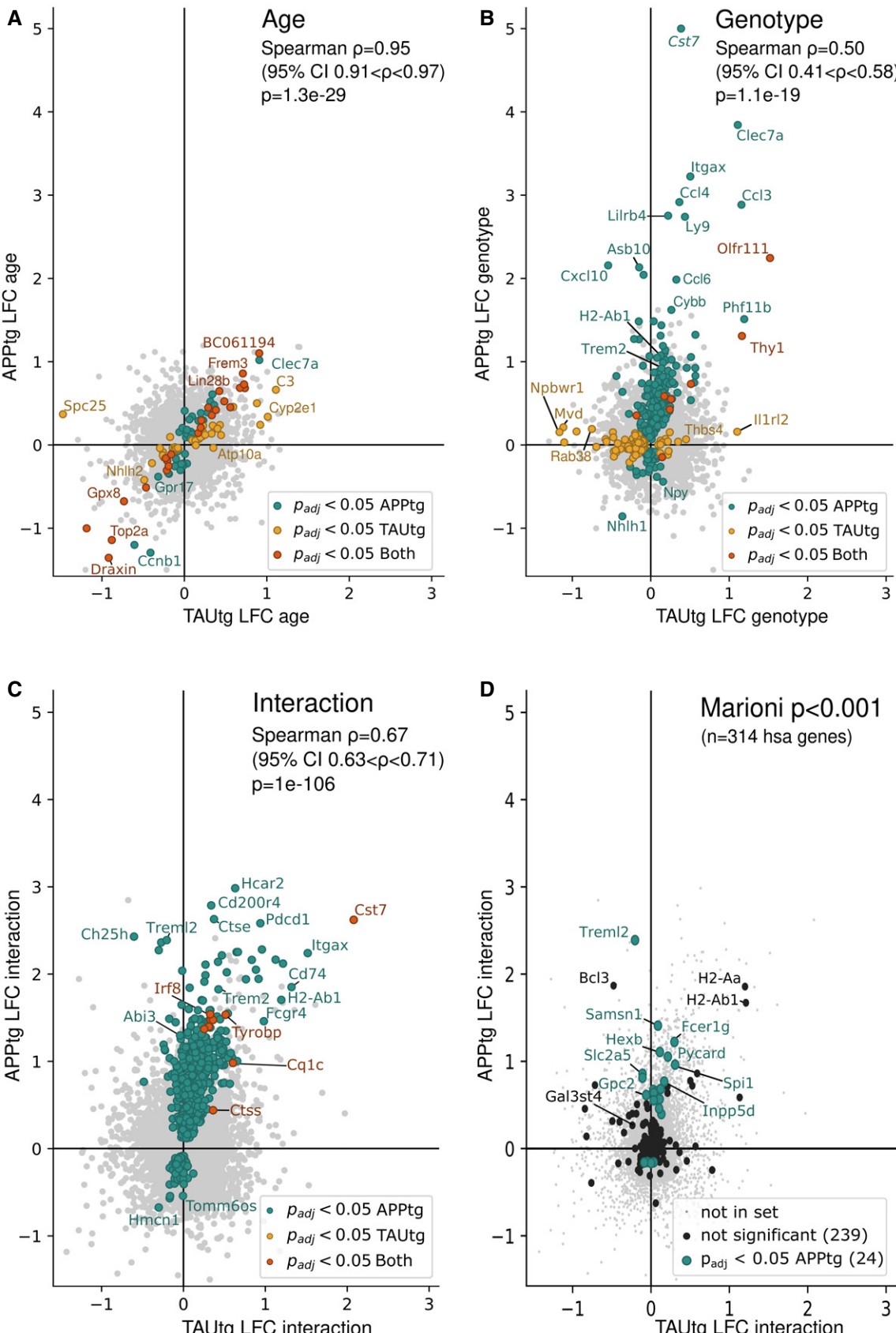

Figure 2.

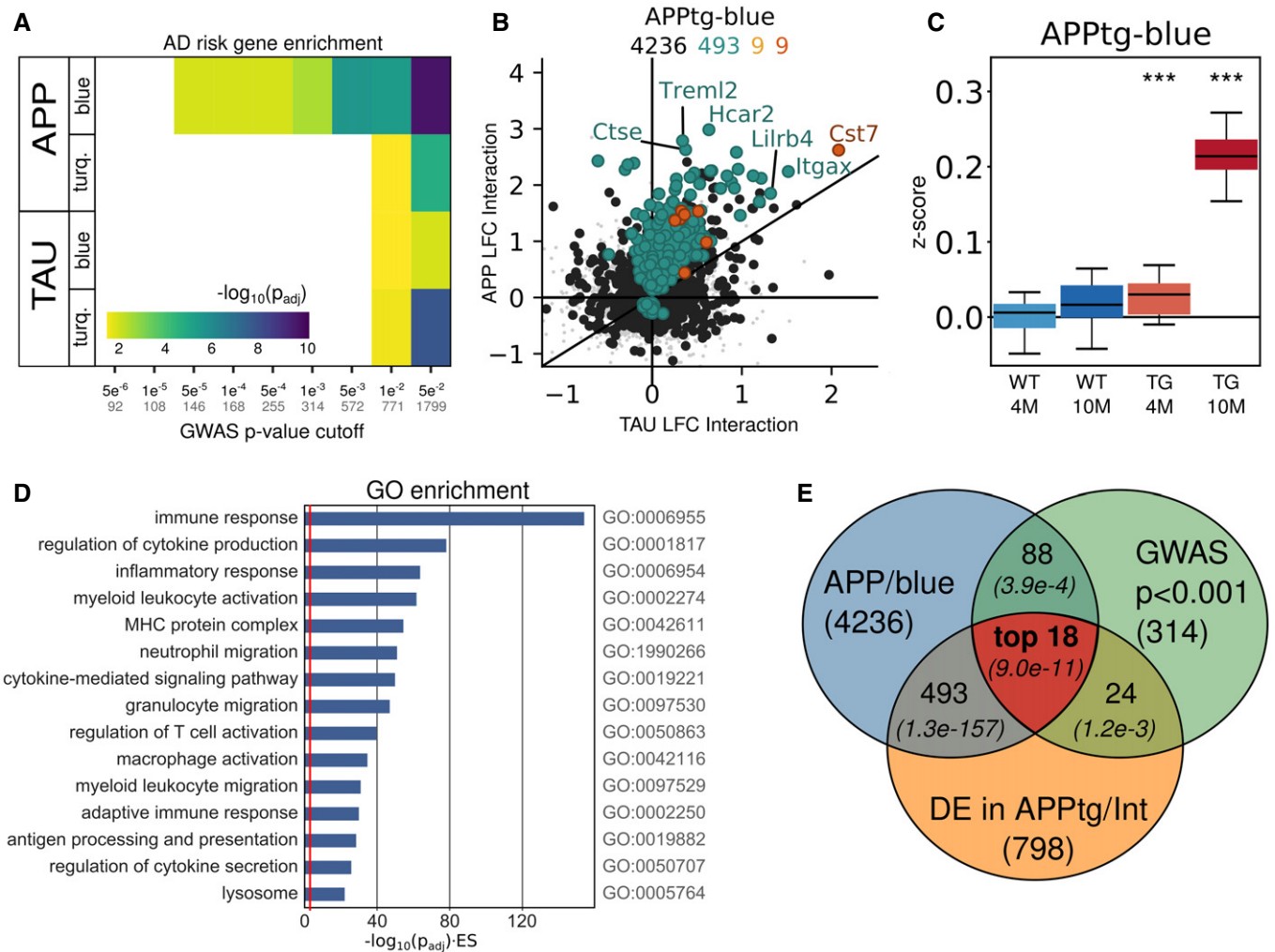

**Figure 3. The APPtg-Blue module represents a coordinated transcriptional response only present in APPtg mice.**

A   Gene set enrichment analysis using Fisher's exact test of GWAS genes from Marioni *et al* (2018) at different *P*-value cut-offs among the WGCNA-derived modules in APPtg or TAUtg mice (see also Appendix Figs S2 and S3). Colors represent −log10 (Benjamini–Yekutieli-adjusted *P*-value) for the enrichment. Numbers on the *x*-axis: black = *P*-value cut-off; gray = size of GWAS gene set.

B   Log2 fold change (LFC) in TAUtg (*x*-axis) and APPtg mice (*y*-axis) after differential expression analysis, assessing the effects of age*genotype interaction. Color code for dots/numbers: gray = genes in hippocampus (n = 15,824); black = genes in APPtg-Blue (n = 4,236); green = genes significantly differentially expressed in APPtg mice (n = 493); yellow = significantly differentially expressed genes in TAUtg mice (n = 9); and red = significantly differentially expressed genes in both (n = 9).

C   *Z*-score distribution per experimental group for all genes within the APPtg-Blue module. Boxplots: center line, median; box limits, 25th–75th quartiles; and whiskers, 1.5× interquartile range. Empirical *P*-values are Bonferroni-adjusted (***$P_{bonf}$ < 0.001) and indicate significant shift in *z*-score distribution (see Materials and Methods).

D   Gene Ontology (GO) enrichment for genes within the APPtg-Blue module. The *x*-axis depicts −log10(FDR-adjusted *P*-value) multiplied by the enrichment score (ES), where the red line represents an −log10(0.049)*(ES = 1).

E   The "top 18" GWAS genes are prioritized by finding the intersection of genes within the APPtg-Blue module (n = 4,236), AD GWAS genes with *P* < 0.001 in Marioni *et al* (2018) (n = 314), and significantly differentially expressed genes in the age*genotype comparison of APPtg mice (n = 798), using the SuperExactTest (Wang *et al*, 2015). Numbers in italics are the Benjamini–Yekutieli-adjusted *P*-values for finding overlap between the different gene sets (see Appendix Table S3).

We identified transcription factors potentially regulating this module using i-CisTarget (Imrichová *et al*, 2015, see also Appendix Fig S2). Out of 16 APPtg-Blue-associated transcription factors, *Spi1* (a.k.a. PU.1), which is a determinant of myeloid fate, comes out as the top candidate, along with other microglia-related and interferon-responsive transcription factors Stat2, Stat1, Ets1, and Irf7 (see Fig 4C). Both *Spi* and *Stat1* are significantly differentially expressed in the APPtg age*genotype comparison (*Spi1* LFC:

0.96, $P_{adj}$ = 9.92e-05; *Stat1* LFC: 0.39, $P_{adj}$ = 0.0013). To summarize, we can conclude that the APPtg-Blue module shows a coordinated transcriptional response to increasing Aβ load employing a large number of AD risk genes. This cellular response involves microglial and astrocyte genes and seems, at least partially, regulated by the transcription factor *Spi1*.

Of note, no such module enriched for AD risk genes is found in the TAUtg mice (see Fig 3A). Several smaller modules show

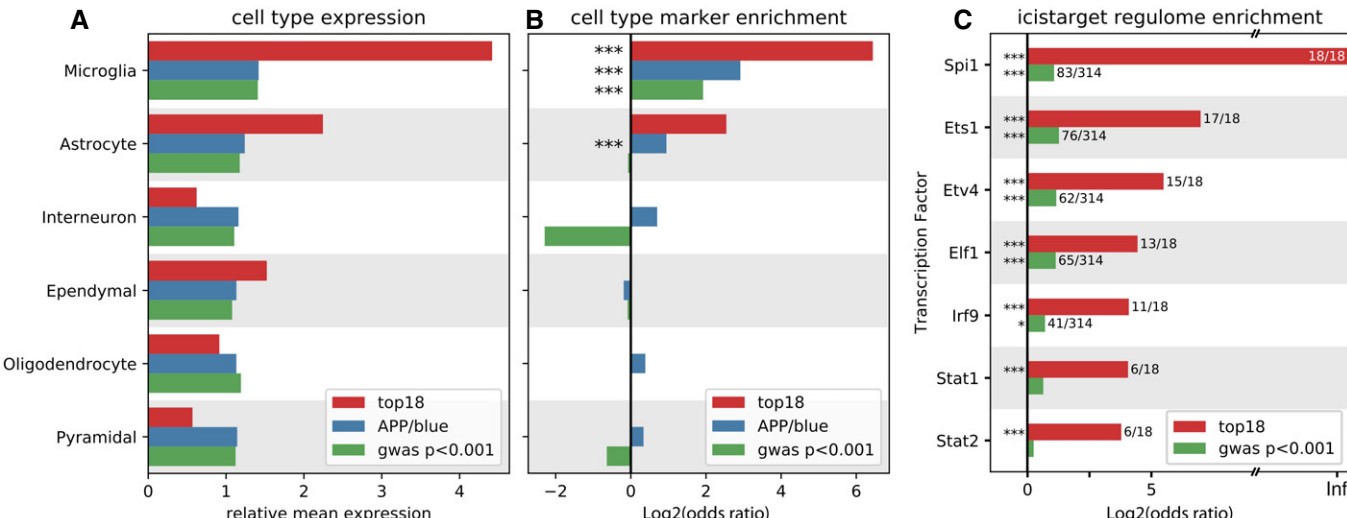

**Figure 4. "Top 18" genes are expressed in microglia and regulated by *Spi1*.**

A For each gene set (top 18, APPtg-Blue, and GWAS *P* < 0.001, Marioni *et al* (2018)), their expression was assessed in the different cell types listed on the left, based on the expression matrix as published by Zeisel *et al* (2015). As can be viewed, the top 18 genes are highly expressed in microglia.

B Based on the marker genes for each cell type as determined by Zeisel *et al* (2015), enrichment of these marker genes was assessed among the three gene sets (top 18, APPtg-Blue, and GWAS *P* < 0.001, Marioni *et al*, 2018). Significant enrichment of microglial marker genes can be observed for all three gene sets, while the APPtg-Blue module is also enriched for astrocyte marker genes.

C Enrichment of transcription factor targets among the "top 18" genes (red bars) and the GWAS genes with *P* < 0.001 (green bars), using i-CisTarget (Imrichová *et al*, 2015).

Data information: Significant enrichment: ****P* < 0.001; **P* < 0.05. *P*-values are Benjamini–Yekutieli-adjusted.

significant overlaps with the microglial gene set (TAU-Red LOR: 2.06, $P_{adj}$ = 5.33e-35; TAU-Paleturquoise LOR: 2.14, $P_{adj}$ = 0.013), the astrocyte gene set (TAU-Red LOR: 1.10, $P_{adj}$: 3.8e-03), and the pyramidal neuron gene set (e.g., TAU-Black LOR:1.28, $P_{adj}$:7.64e-12 and TAU-Pink LOR:1.02, $P_{adj}$: 3.6e-03; see Appendix Figs S2 and S3 and Appendix Table S2), suggesting that the response to Tau pathology is also multicellular by nature. However, the fragmented cellular response to TAU pathology is clearly different from the coordinated transcriptional regulation response in the APPtg-Blue module.

**Prioritization of subthreshold GWAS risk genes of AD within the Aβ-induced transcriptional response**

The set of 1,799 GWAS genes ($P_{mar}$ < 0.05) contains 439 genes co-regulated within the APPtg-Blue module, further demonstrating that this constitutes a core pathobiological response to Aβ onto which genetic risk for sporadic AD converges. These 439 risk genes encompass well-established ("canonical") AD genes, i.e., *Abi3, Apoe, Bin1, Clu, Def8, Epha1, Fcer1 g, Fermt2, H2-Ab1* (*HLA-DQA* in humans), *Inpp5d, Plcg2, Prss36, Rin3, Siglech* (*CD33* in humans), *Spi1, Tomm40, Trem2,* and *Zcwpw1*. The 421 other genes in this data set have been associated to AD with decreasing degrees of statistical certainty. When taking gene sets with increasingly stringent *P*-values for their association to AD risk, the number of GWAS genes in the selection obviously drops (Appendix Table S1), but the genetic evidence for the role of these genes in AD increases. In the end, the choice of the *P*-value threshold is somewhat arbitrary, as even SNPs with *P*-values up to 0.5 contribute to the PRS and

therefore contain information (Escott-Price *et al*, 2017). The important insight is, however, that identified genes are implied in AD by two independent unbiased experimental approaches here: genetically, by contributing to the risk of AD in human studies, and functionally, by being part of the APPtg-Blue response module in a mouse model for AD (Escott-Price *et al*, 2017; Marioni *et al*, 2018). We focus here on GWAS genes that (i) have been associated to AD with $P_{mar}$ < 0.001, as these genes are more likely to make an individual impact on disease, (ii) that are present in the APPtg-Blue module, and (iii) that are, additionally, significantly differentially expressed in the age*genotype comparison of APPtg mice (see Fig 3E). The latter criterion is restrictive because we require that the gene is not only part of the APPtg-Blue response module but is also significantly altered in expression, thus part of the core response that defines the module. These three criteria show a highly significant intersection of 18 genes which is markedly better than expected by chance alone (expected 2.2 genes, $P_{adj}$ = 9.0e-11, SuperExactTest; Wang *et al*, 2015, Fig 3E and Appendix Table S3). These 18 genes are *APOE, CLU, GPC2, INPP5D, CD33, PLCG2, TREML2, SPI1, FCER1G, SYK, GRN, SLC2A5, SAMSN1, PYDC1, HEXB, RRBP1, LYN,* and *BLNK* (see Fig 5). The full set of GWAS genes with $P_{mar}$ < 0.5 and significantly differentially expressed in the APPtg-Blue module (*n* = 263 genes) are listed in Dataset EV4.

Among the 18 genes, we find 7 established AD GWAS variants, i.e., reaching genome-wide significance in one or more studies (Taguchi *et al*, 2005; Efthymiou & Goate, 2017; Verheijen & Sleegers, 2018). We indicate them here together with the Bonferroni-adjusted *P*-value ($P_{mar}$) for the GWAS association as listed in Marioni *et al* (2018): *APOE* ($P_{mar}$: 2.80e-118), *CLU* ($P_{mar}$: 7.68e-15),

*INPP5D* aka SHIP1 ($P_{mar}$: 2.42e-10), *CD33* (Siglech in mice; $P_{mar}$: 1.82e-07), *PLCG2* ($P_{mar}$: 2.76e-06), *SPI1* ($P_{mar}$: 5.45e-06), and *FCER1G* ($P_{mar}$:1.10e-05). The other 11 genes have not reached genome-wide statistical significance in any previously published GWAS study but are prioritized here based on additional significant observations, i.e., that these genes are significantly differentially expressed when exposed to pathology and are present in the APPtg-Blue module as discussed. Although each of these observations has an associated *P*-value, they are derived from different hypotheses, making it difficult to combine them into a specific *P*-value per gene. However, by using an extension on the classical Fisher's exact test (SuperExactTest; Wang *et al*, 2015) to evaluate the significance of finding overlaps between multiple gene sets, it was demonstrated that observing an intersection of 18 genes among these three gene sets is highly significantly better than expected by chance ($P_{adj}$ = 9.0e-11, see Appendix Table S3 and Fig 3E).

Among these 11 novel genes (their individual $P_{mar}$ is again indicated), *SYK* ($P_{mar}$: 1.41e-04), *LYN* ($P_{mar}$: 8.38e-04), and *BLNK* ($P_{mar}$: 9.52e-04) together with the previously mentioned "canonical" genes *INPP5D*, *FCER1G*, and *PLCG2* play a role in FC gamma receptor-mediated phagocytosis (see also Fig 5). When examining the longer list of priority GWAS genes (see Dataset EV4), we find more members of this pathway, including *TREM2*, *ITGAM*, and *VAV1*. Moreover, the same genes, as well as others of the top 18 list, i.e., *PYDC1/PYCARD* ($P_{mar}$: 4.69e-04), *GRN* ($P_{mar}$: 1.56e-04), *SLC2A5* ($P_{mar}$: 2.07e-04), and *SAMSN1* ($P_{mar}$: 4.66e-04), are all part of the "immune and microglia" module regulated by *TYROBP* inferred by Zhang *et al* from RNA-seq data derived from late-onset AD patients (Zhang *et al*, 2013). Other microglia- and lysosome-related genes are *TREML2* ($P_{mar}$: 4.90e-06) and *HEXB* ($P_{mar}$: 4.87e-04). *RRBP1* ($P_{mar}$: 7.10e-04) is an ER-associated ribosomal protein (Savitz & Meyer, 1993), with high expression in glial cells (Dataset EV4). All of the 18 AD risk genes are expressed in microglia according to the expression data from Zeisel *et al* (see Fig 4A; Zeisel *et al*, 2015) and are predicted *SPI1* targets according to i-cisTarget (see Fig 4C), and 11 out of these (*APOE*, *BLNK*, *HEXB*, *INPP5D*, *LYN*, *PLCG2*, *RRBP1*, *SAMSN1*, *SLC2A5*, *SPI1*, *SYK*) are demonstrated *SPI1* targets in a ChIPseq experiment in the BV2 microglia cell line (Satoh *et al*, 2014). The possible role of these genes together with other microglia-related previously established GWAS genes is schematically summarized in Fig 5. In conclusion, many of these subthreshold AD risk genes that contribute to polygenic inheritance participate in microglia-related functional networks both in AD patients and in mouse models of Aβ pathology.

## Single microglia sequencing confirms high proportions of activated microglia in APPtg mice

Given that the neuroinflammatory and microglial response appears markedly different in APPtg and TAUtg mice, we performed single cell sequencing on FACS-sorted CD11b$^+$/CD45$^+$ hippocampal microglia on all 8 experimental groups, with the minor modification that the oldest mice were 11M instead of 10M. After filtering and doublet removal, 15,599 single cells remained, which were subjected to dimensional reduction, batch effect removal, and clustering with Seurat (Butler *et al*, 2018; Stuart *et al*, 2019). Nine different clusters were obtained (see Fig 6A) and, after matching the expression of their marker genes with publicly available gene sets

(see Appendix Fig S4; Friedman *et al*, 2018; Mancuso *et al*, 2019a; Sala Frigerio *et al*, 2019), were termed activated response microglia (ARM, *n* = 2,841 cells), CNS-associated macrophages (CAMs, *n* = 720), cycling and proliferating microglia (CPM, *n* = 285, see Dataset EV6 for a list of marker genes), homeostatic microglia cluster 1 (HM.1, *n* = 7,111), homeostatic microglia cluster 2 (HM.2, *n* = 2,895), interferon-response microglia (IRM, *n* = 387), high major histocompatibility complex-expressing microglia (MHC.high, *n* = 144, see Dataset EV6 for a list of marker genes), monocytes (Mnc, *n* = 100), and transitioning response microglia (TRM, *n* = 1,115). As described in Sala Frigerio *et al* (2019), HM.2 cells appear to be in a slightly more metabolically and transcriptionally active state than HM.1 cells, with mild upregulation (LFC: +0.1 to +0.6) of *Apoe*, *Lyz2*, *H2-D1*, and *Cd52* and many ribosomal and mitochondrial genes (see Dataset EV5).

When assessing the distribution of cells of each experimental group over the different clusters, it becomes apparent that the ratio of HM (HM.1 and HM.2) versus ARM is vastly different among the experimental groups (Fig 6B and C, Appendix Fig S5). Whereas WT mice from both strains and ages have 4–10% of their cells classified as ARM, TAUtg show mild increases to 14.9 and 21.3% at 4M and 11M, respectively, but APPtg show drastic increases in the number of ARMs with aging, from 26.5% at 4M to 56.7% at 11M. Although only few IRMs can be observed, APPtg mice have almost twice the percentage of cells in this cluster (4M: 5.2%; 11M: 5.9%) compared to the TAUtg mice (4M: 2.2%; 11M: 3.2%) and up to 9 times the percentage of WT mice (0.7–2.3%).

When placing all microglial cells on a pseudotime trajectory using Scorpius (Fig 6D, preprint: Cannoodt *et al*, 2016), it becomes clear that WT microglia stay within the homeostatic state. In contrast, APPtg-4M mice show already an initial shift of phenotype from homeostatic toward activated, yet in APPtg-11M mice, microglia appear to become even more activated and the ratio of HM versus ARM clearly shifts toward ARM. In TAUtg mice, however, such an extreme phenotypic shift is not observed, with few ARM cells reaching the final stage of activation in TAUtg-11M mice (Fig 6D).

When comparing all ARM cells to all HM cells within the APPtg and the TAUtg mice, and using Spearman correlation with the results from the differential expression (Fig 6E), it is clear that the overall ARM phenotype between APPtg and TAUtg mice is not different (Spearman correlation *R* = 0.91, *P* = 2.2e-16; see also Dataset EV7), with increased expression of *Apoe*, *Ccl4*, *Ccl3*, *Clec7a*, and *Lyz2*, and downregulation of homeostatic markers *P2ry12*, *Tmem119*, and *Nav2*. Thus, whereas the ARM profile in TAUtg or APPtg mice is not intrinsically different, Aβ exposure appears to induce a more rapid and severe ARM response. Despite many early changes in transcription, the inflammatory response in the TAUtg mice appears late and as a relatively mild part of the pathology.

## Microgliosis mainly contributes to the expression of top 18 genes

When assessing the top 18 genes within the single microglia dataset, we can observe very low expression of *Clu*, *Gpc2*, and *Treml2* and modest expression of *Syk* and *Plcg2* (see Fig EV2). With cut-offs of $P_{adj}$ < 0.05 and −0.1 < LFC >+0.1, we demonstrate that *Slc2a5*,

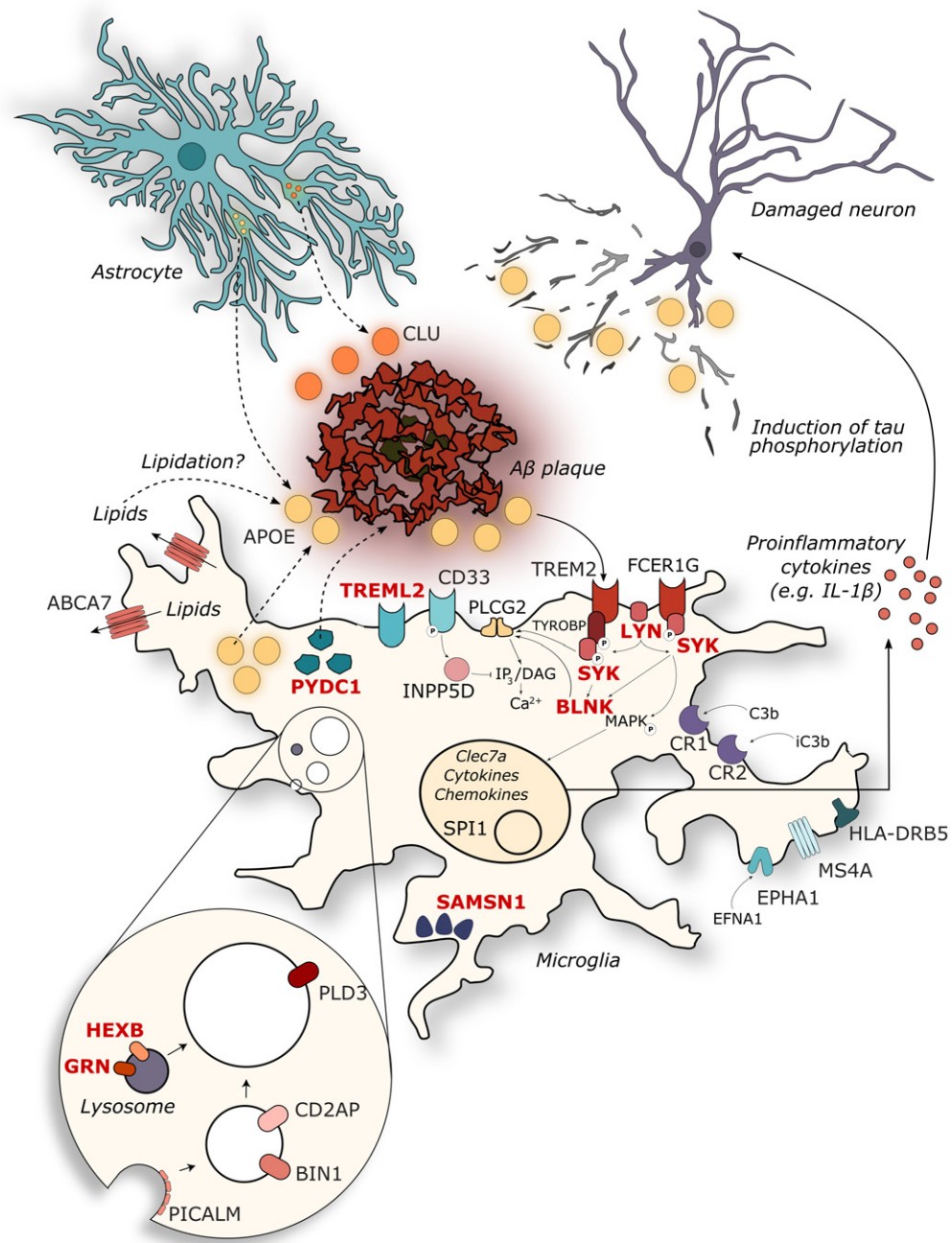

**Figure 5.  The 11 novel prioritized AD risk genes are within a lysosomal gene networks or involving innate immunity.**

Genes are depicted in their known cellular components, with previously described established GWAS (Efthymiou & Goate, 2017; Verheijen & Sleegers, 2018) in black and the prioritized novel genes (this paper) in red. See discussion section for further functional annotation and references.

*Siglech, Inpp5d, Pycard, Hexb, Lyn, Spi1,* and *Rrbp1* are all significantly higher expressed in HM compared to ARM, whereas *Fcer1g* and *Apoe* are significantly higher expressed in ARM compared to HM. The enrichment for the top 18 genes among genes upregulated in HM cells compared to ARM cells was found significant (Fisher's exact test, $P = 7.0e\text{-}4$). Publicly available data (http://research-pub. gene.com/BrainMyeloidLandscape/reviewVersion/; Keren-Shaul *et al*, 2017; Sala Frigerio *et al*, 2019; preprint: Srinivasan *et al*,

2019) corroborate that most of our top 18 genes are significantly upregulated in bulk RNA-seq from PS2APP mice and AD patient cortical samples, but remains unchanged or slightly downregulated in RNA-seq data from isolated microglia (see Fig EV2).

We aimed to further understand for each of the top 18 genes what the microglial contribution was to the observed increased expression in the bulk RNA-seq data. A regression model was used to assess how well the bulk RNA-seq expression data for a specific

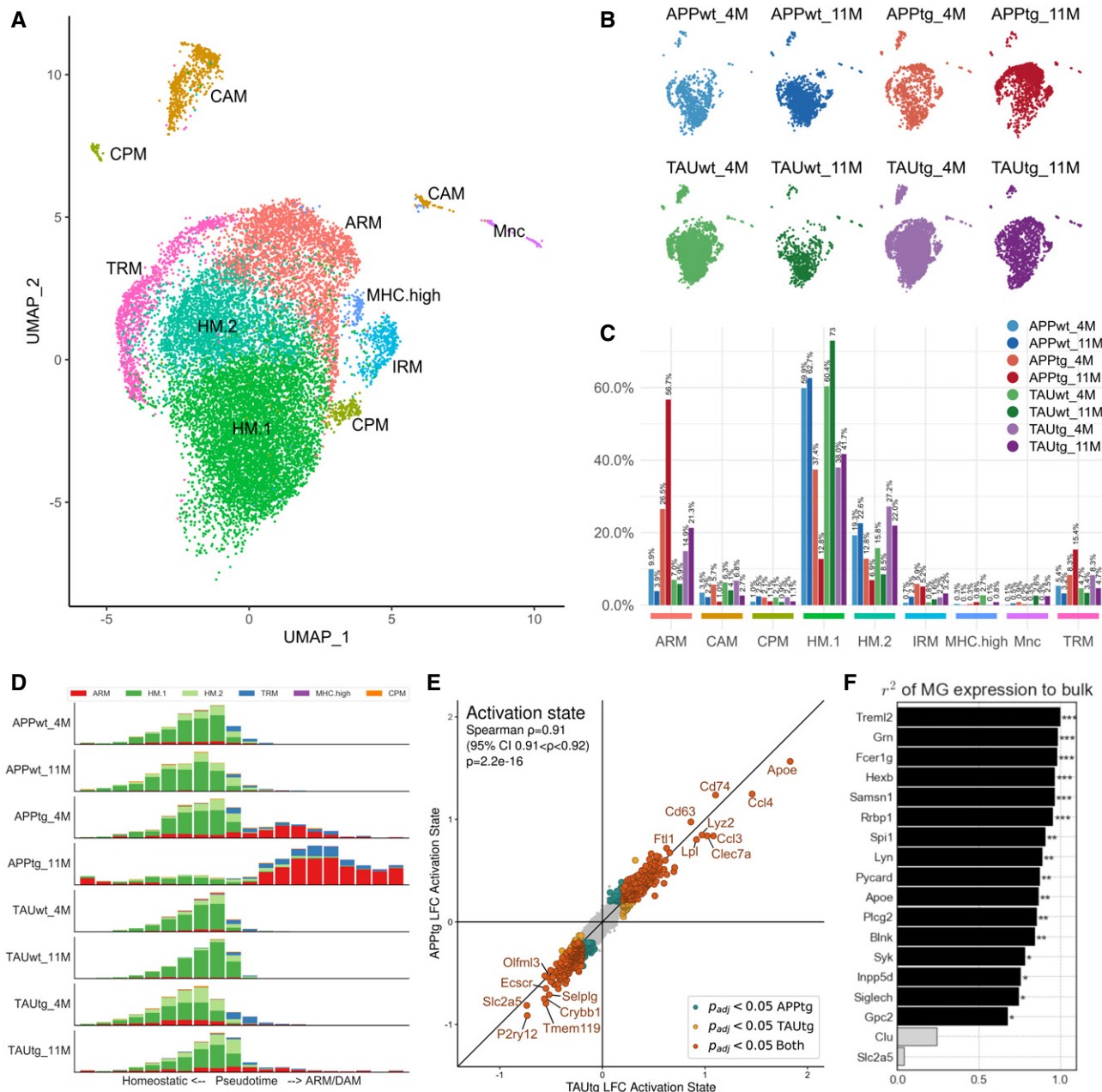

**Figure 6. Single microglia sequencing demonstrates a strong ARM response in APPtg in comparison with TAUtg mice.**

A  UMAP clustering of sorted Cd11b⁺/Cd45⁺ myeloid cells from mouse hippocampus. ARM: activated response microglia; CAM: CNS-associated macrophages; CPM: cycling and profilerating microglia; HM.1: homeostatic microglia cluster 1; HM.2: homeostatic microglia cluster 2; IRM: interferon-response microglia; MHC.high: high MHC-expressing microglia; Mnc: monocytes; TRM: transitioning microglia.

B  Distribution of cells across the different clusters per experimental group (*n* = 2 mice pooled for all experimental groups, except for TAUtg-11M and TAUwt-11M with *n* = 3 mice pooled).

C  Distribution of cells per experimental group over the different clusters, expressed in percentages as stated on top of each bar. E.g., of all cells within APPtg-11M, 56.7% of cells are ARM, while 12.5% are HM.1 cells.

D  Scorpius-based ranking of all cells on a pseudotime from homeostatic to activated microglia. Whereas all cells within the WT groups are clustered within the homeostatic part of the pseudotime trajectory, APPtg-4M and APPtg-11M mice demonstrate a shift toward more activated microglia.

E  Differential expression of ARM versus HM.1 microglia for APPtg and TAUtg mice separately. Genes with a positive log(fold change; LFC) are more highly expressed in ARMs compared to HM.1 microglia; genes with a negative LFC are more highly expressed in HM.1 compared to ARM cells. As can be seen genes that are significantly differentially expressed between ARM and HM.1 cells are highly similar among APPtg and TAUtg mice (Spearman correlation r = 0.91, P = 2.2e-16). LFC > |0.2| with $P_{adj}$ < 0.05 is considered significant.

F  Predicted contribution of microglial expression of each of the top 18 genes to the observed bulk RNA-seq expression data, based on linear regression (see Materials and Methods). For 16/18 genes, the microglial expression contributes significantly and 67% or more to the observed expression from the bulk RNA-seq. ***Benjamini–Yekutieli-adjusted P-value ($P_{adj}$) < 0.001; **$P_{adj}$ < 0.01; *$P_{adj}$ < 0.05.

gene could be explained by a combination of the microglial expression of that gene in homeostatic and non-homeostatic microglia and the extent of microgliosis for each microglial cell state. Using the regression coefficient $R^2$ and the Benjamini–Yekutieli-adjusted *P*-value, we conclude that the bulk RNA-seq expression pattern for *Clu* and *Slc2a5* cannot be explained by the expression of these genes in microglia ($P_{adj} < 0.05$, see Fig 6F). For the other 16 of the top 18 genes, the observed variation in expression in the bulk RNA-seq can be explained by the predicted variation in expression in microglia for 67% or more. By looking more closely at the expression pattern of specific genes among the different microglia cell states, it becomes clear that the upregulation of expression reflects two different responses. As expected, the ARM response explains upregulation of *Apoe, Fcer1g,* and *Treml2,* (see Fig EV3). Other genes are more highly expressed in HM cells (e.g., *Inpp5d* and *Siglech*), and thus, the mere shift toward the ARM cell state is insufficient to explain their increased expression in the bulk RNA. We conclude that for these genes the microgliosis is the major driver of the gene upregulation (see Fig EV3). Thus, based on these predictions we can conclude that the increased expression in the bulk RNA-seq of our top 18 genes is driven both by a shift in cell state but also by increased number of microglia cells. This suggests that two microglia responses contribute to the global inflammatory phenotype in the APPtg mice.

## Discussion

This study identifies a gene transcription module (APPtg-Blue) that is specifically induced by overexpression of mutated *APP/PSEN1* (but not by *MAPT*) and is significantly enriched for a large set of genes that have previously been linked with SNPs associated to AD risk without reaching genome-wide significance (Marioni *et al*, 2018). Our experiments suggest that these genes are part of one or more pathways that characterize the microglial response to Aβ (see Fig 5). While the individual effects of various SNPs on expression of these genes are likely small and biologically insignificant, the combination of hundreds of such subtle alterations, if synergic in one pathway, might veer the cellular response to Aβ in a disease-causing direction through biological epistasis (Moore & Williams, 2005).

Functionally, this co-regulatory network (the APPtg-Blue module) represents a large neuroinflammatory response involving astroglia and, overwhelmingly, microglia. These observations considerably expand previous work, strengthening the hypothesis that the major response in the brain to Aβ pathology is neuroinflammation (Wirz *et al*, 2013; Matarin *et al*, 2015). Our and other data suggest that an early and profound neuroinflammatory response is an integral and perhaps even driving component in AD pathogenesis and that Aβ is sufficient to induce this innate immune reaction (Zhang *et al*, 2013; Matarin *et al*, 2015).

Depending on the TAU mouse model used, others have shown that microglia play an important role in neurodegeneration caused by human mutant TAU (Shi *et al*, 2017, 2019; Mancuso *et al*, 2019b) and that the ARM (or DAM) response is similar in APP and TAU mouse models (Friedman *et al*, 2018). However, these particular TAU mouse models appear to develop a very aggressive phenotype, with motor and cognitive deficits developing earlier [already

from 2.5 months of age (Allen *et al*, 2002; Yoshiyama *et al*, 2007; Scattoni *et al*, 2010)] and loss of synaptic integrity and microglial activation preceding tangle formation (Yoshiyama *et al*, 2007) compared to the TAUtg model used here. We speculate that the DAM response at those stages of disease might reflect responses to dying neurons which are known to induce a similar response in microglia (Deczkowska *et al*, 2018). Despite the cognitive alterations in the TAUtg model in this study, overall neuronal loss at 10M remains rather mild (up to 35% in the CA1 at 12M in TAUtg mice; Schindowski *et al*, 2006). The TAUtg mice used here thus differ from other models of TAU pathology (Allen *et al*, 2002; Yoshiyama *et al*, 2007; Scattoni *et al*, 2010) where tangles appear very rapidly and cause massive neuronal loss. We suggest that the severe cell death in these other models leads to a strong microglial response, opposed to our model where this response remains mild. Heneka and colleagues also found in the same TAUtg model that the inflammatory response was not genotype-dependent and in fact was highly similar to the aging phenotype in WT controls, which is in line with our conclusion that the intraneuronal TAU pathology does not induce massive inflammatory reactions (Ising *et al*, 2019). Thus, the microglia responses recorded in these other TAU models are clearly not a direct response to the TAU pathology as such, as similar Tau-pathology is also obviously present in our model (see Fig 1A; Schindowski *et al*, 2006; Ising *et al*, 2019), but instead to the cellular damage caused downstream of the TAU insult.

The microglia response in the APPtg mice is already present at 4 months, even before cognitive alterations appear. Thus, in contrast to the TAUtg mice, here the microglia seem to provide a direct response to the rising Aβ pathology, with inflammation becoming massive at the late stage of the disease.

In AD patients, it remains to be clarified where, when, and how Aβ and TAU pathology relate to each other and how these contribute to disease progression toward dementia. Our data, however, suggest that Aβ drives a very early and strong neuroinflammatory response (APPtg-Blue module) while TAU pathology (at least in our mild model) is characterized early on by neuronal dysfunction. Recent data suggests that microglia actively participate in the formation of TAU-related pathologies, including TAU-positive dystrophic neurites and TAU-induced neurodegeneration (Shi *et al*, 2017, 2019; Mancuso *et al*, 2019b; Prokop *et al*, 2019). Our hypothesis implies that TAU pathology comes downstream of the microglia/neuroinflammatory response on Aβ. The TAUtg transcriptional analysis does suggest, however, that TAU induces very early downregulation of neuronal genes (Fig EV1), which vibes with the observations that dementia correlates much better with the appearance of tangles than amyloid plaques. Besides its role in regulating immune response, the newly prioritized risk gene *Syk* is a known TAU kinase, providing an interesting potential link between Aβ and TAU pathology (Paris *et al*, 2014).

Earlier studies have relied heavily on functional analysis and module connectivity for putative gene selection involved in AD pathogenesis, but whether these genes were also differentially expressed was not considered (Zhang *et al*, 2013; Matarin *et al*, 2015; Salih *et al*, 2019). We used two independent functional criteria: Genes must have a genetic association with AD at $P < 0.001$ and show a statistically significant change in expression when exposed to increasing levels of Aβ over time, indicating that these genes are part of the cellular response. Therefore, we can predict with high

confidence (BY-$P_{adj}$ = 9.0e-11) that they are indeed involved in an AD-relevant disease pathway responding to Aβ pathology. It must be noted that the GWAS genes currently used were based on genomic proximity to AD-associated SNPs, which may result in spurious associations. Although this potential caveat may affect our outcome, it is unlikely that the expression of such an erroneously identified gene would be significantly altered in AD-relevant pathways and thus would appear in our final list.

Among the prioritized SNPs, many are linked to known myeloid-expressed genes involved in innate immune pathways (see Fig 5). Briefly, *FCER1G* encodes for the gamma subunit of IgE-specific Fc receptors and plays a role in phagocytosis and microglial activation (Nimmerjahn & Ravetch, 2008). *SYK* is recruited by *FCER1G* and *TYROBP* (DAP-12) upon activation of *TREM2*, whereas *LYN* is localized at the plasma membrane and mediates *TREM2/SYK* phosphorylation (Kipps *et al*, 2017; Konishi & Kiyama, 2018). *BLNK* is a target of *SYK* and mediates the recruitment of *PLCG2* (Kipps *et al*, 2017). *PYCD1* (*Pycard* in mouse) encodes for apoptosis-associated speck-like protein containing C-terminal caspase recruitment domain (ASC), a core component of the inflammasome complex, and has been linked to seeding and spreading of Aβ in mouse models of AD (Venegas *et al*, 2017). The lysosomal protein *HEXB* and glucose transporter *SLC2A5* are part of the microgliome profile (Butovsky *et al*, 2014). *GRN* can adopt multiple subcellular localizations from lysosomes (as granulin) to the extracellular space (as progranulin) and exerts a negative control upon microglial activation in mouse models (see Fig 5; Martens *et al*, 2012). Interestingly, all of the prioritized AD risk genes are *Spi1* targets, substantiating the pivotal role of this transcription factor in the observed Aβ-induced transcriptional response (Gjoneska *et al*, 2015; Huang *et al*, 2017). The single microglia sequencing data demonstrate that 8 of the 18 prioritized genes are more highly expressed in homeostatic microglia (HM) than in activated microglia (ARM). Using our regression model to predict the contribution of microglia to the observed bulk RNA-seq expression profile, we could demonstrate that for the top 18 genes most of the increased expression in the bulk RNA-seq could be explained by microgliosis. It remains to be elucidated whether the loss of homeostasis or the acquisition of an activated microglial phenotype is ultimately more detrimental to pathology progression.

In conclusion, we suggest that the transition from a rather asymptomatic biochemical or prodromal phase, to the clinical, symptomatic, phase in AD, is largely determined by the combined inheritance of low-penetrant SNPs, which, as our work suggests, might affect the gene expression in the APPtg-Blue module. The concept of PRS forces us to abandon the classical gene-based view on disease etiology and to consider GWAS genes as part of a functional network (van der Sijde *et al*, 2014). Given the robustness of biological systems and (the theory of) genetic buffering, a single SNP within the network will not lead to disease (Moore, 2005). However, multiple SNPs within the same network may tip the balance to a disease-causing disturbance. Such hypothesis also provides an explanation for the conundrum that some patients with high Aβ burden succumb without clinical symptoms (Chételat *et al*, 2013). While Aβ might be the trigger of the disease, it is the genetic make-up of the microglia (and possibly other cell types) which determines whether a pathological response is induced. Identifying which SNPs are crucial to such network disturbances and how

those SNPs lead to altered gene expression will be the next big challenge.

# Materials and Methods

### Study design

The goal of the current study was to assess the enrichment of AD GWAS genes among the transcriptional responses as APPtg and TAUtg mice develop pathology over time. Given the mice' age and genotype, mice could not be distributed randomly across the experimental groups, but randomization was performed during RNA extraction and library preparation to avoid batch effects. Library preparation was performed in a blinded fashion. Details regarding sample size, biological replicates, and outliers can be found below.

### Mice

The local Ethical Committee of Laboratory Animals of the KU Leuven (governmental license LA1210591, ECD project number P202-2013) approved all animal experiments, following governmental and EU guidelines. APPtg (a.k.a. B6.Cg-Tg; Thy1-APPSw,Thy1-PSEN1*L166P; 21Jckr; Radde *et al*, 2006) mice express $APP^{Swe}$ and $PSEN1^{L166P}$ transgenes, while TAUtg (a.k.a. THY-Tau22 or Tg; Thy1-MAPT; 22Schd; Schindowski *et al*, 2006) mice express the 412 aa isoform of the human 4-repeat *MAPT* gene containing the G272V and P301S mutations. Both mice are made using the *Thy1.2* promoter. Mice were group housed (max. 5/cage), on a 12-h day/night cycle, with *ad libitum* access to food and water. Littermate transgenic (TG) and wild-type (WT) male mice were sacrificed at 4 months (average 124 days, SD 1.84 days), 10 months of age (average 300 days, SD 2.22 days), or 11 months of age (average 337 days, SD 2.78 days), creating 8 experimental groups ($n = 12$ per group for bulk RNA-seq; see Fig 1A; $n = 2$–3 per group for single microglia sequencing). For bulk RNA-seq, number of mice was determined as described in Sierksma *et al* (2018), and following cervical dislocation, 4M and 10M mice hippocampi were microdissected, snap-frozen in liquid nitrogen, and stored at −80°C. For single microglia sequencing, 4M and 11M mice received an overdose of sodium pentobarbital and were perfused with heparinized PBS, and their hippocampi were microdissected.

### Immunohistochemistry

Mice of 9 months of age were sacrificed for immunohistochemistry (APPtg: 3 males; TAUtg: 3 males; TAUwt: 2 females; APPwt: 1 male) through an overdose of sodium pentobarbital and transcardial perfusion of PBS followed by 4% paraformaldehyde. Brains were cut using a vibratome into 30 μm thick coronal sections. Sections were permeabilized in 0.2% Triton X100 PBS (PBS-T) for 15 min and subsequently incubated with X34 staining solution [10 μM X34 diluted in 60% PBS (vol/vol), 40% ethanol (vol/vol), and 20 mM NaOH, see (Ulrich *et al*, 2018)]. X34 (SML1954-5MG, Sigma-Aldrich) is a fluorescent Congo Red derivative (Styren *et al*, 2000), binding β-sheet secondary protein structures, thus detecting both Aβ plaques and TAU tangle-like inclusions. Sections were then washed 3× for 2 min in 60% PBS (vol/vol) with 40% ethanol (vol/vol) and

then washed 2 × 5 min in PBS with 0.1% Triton X-100. After blocking (5% donkey serum in PBS-T), sections were incubated overnight with the primary antibodies (anti-Iba1 1/500 from Wako (#019-19741) and anti-NeuN 1/300 from Synaptic Systems (#266004)) in 0.5% donkey serum in PBS-T at 4°C. Next, sections were washed in PBS-T and incubated with the secondary antibodies (donkey anti-rabbit Alexa488 (A21206) and goat anti-guinea pig Alexa568 (A11075), both from Invitrogen, 1/500 for 2 h at room temperature) in 0.5% donkey serum in PBS-T, and following additional washes were counterstained with TO-PRO-3 (1/1,000 in PBS, Thermo Fisher Scientific) and mounted with Mowiol solution. Images (z-stacks) were acquired on a Nikon A1R Eclipse Ti confocal microscope.

### RNA extraction, library construction, sequencing, and mapping

For bulk RNA-seq, the left hippocampus of each mouse was homogenized in TRIzol (Invitrogen, Carlsbad, CA, USA) using 1 ml syringes and 22G/26G needles and purified on mirVana spin columns according to the manufacturer's instructions (Ambion, Austin, TX, USA). RNA purity (260/280 and 260/230 ratios) and integrity were assessed using Nanodrop ND-1000 (Nanodrop Technologies, Wilmington, DE, USA) and Agilent 2100 Bioanalyzer with High Sensitivity chips (Agilent Technologies, Inc., Santa Clara, CA, USA) and Qubit 3.0 Fluorometer (Life Technologies, Carlsbad, CA, USA), respectively. RNA integrity values of the samples ranged from 7.9 to 9.3 (median = 8.6).

Library preparation and sequencing were performed at the VIB Nucleomics Core (Leuven, Belgium) using 1 μg of total RNA per sample. Poly-A containing mRNA molecules were purified using Illumina TruSeq® Stranded mRNA Sample Prep Kit (protocol version 15031047 Rev.E) and poly-T oligo-attached magnetic beads. After reverse transcription with random primers, different barcodes were introduced to each sample by ligating a single "A" base to the 3′ ends of the blunt-ended cDNA fragments and multiple indexing adapters and 10 cycles of PCR amplification were performed. Ninety-six libraries were pooled and sequenced in 4 runs using NextSeq 500 High75 Output kits on an Illumina NextSeq 500 instrument (Illumina, San Diego, CA, USA). Reads were pre-processed and mapped using BaseSpace Secondary Analysis (version 2.4.19.6, basespace.illumina.com), filtering out abundant sequences and trimming 2 bp from the 5′ end. Reads were aligned against the mm10/GRCm38 *Mus musculus* reference genome by Tophat2 (version 2.0.7, Trapnell *et al*, 2009).

### Data pre-processing and differential expression analysis (bulk RNA-seq)

We found in 2 of the 12 APPtg-10M mice a 46% lower expression levels of hsa-APPswe, a 26% lower expression of hsa-Psen1L166p, and a 24% reduction in mmu-Thy1 gene, which drives the expression of the transgenes. Principle component analysis (using log2 (raw reads from Feature Counts +1) and the "prcomp" function in R) also revealed that these 2 mice were closer to the 4M APPtg mice than the 10M APPtg mice (see Appendix Fig S1). In the absence of an explanation for this phenomenon, these two mice were excluded from all further analyses.

mRNAs with average raw read counts ≤ 5 in 10 out of 96 samples were discarded, leaving 15,824 mRNAs for differential expression

(DE) analysis. Non-biological variation due to library preparation technicalities (i.e., effects for library prep batch, RNA extraction group, and RNA concentration) was removed by the "removeBatch-Effect" function from limma package 3.22.7 Bioconductor/R (Ritchie *et al*, 2015). DE analysis was conducted using a two-way interaction model (age, genotype, age*genotype, see Fig 1B; two-sided testing) for APPtg and TAUtg mice separately. To adjust for multiple testing, Benjamini–Yekutieli (BY) *P*-value adjustment was performed, as we want to control for false discovery rate across experiments that have partially dependent test statistics; hence, the traditional Benjamini–Hochberg adjustment was not applicable (Benjamini & Yekutieli, 2001). Further differential expression analyses were performed by comparing WT to TG mice at 4M and 10M separately, with BY *P*-value adjustment across all 4 comparisons. Ranking of genes for Spearman correlations (two-sided testing) was always performed on the basis of signed log10(*P*-value), i.e., the log10 of the unadjusted *P*-value with a positive sign if the LFC of that gene within that comparison was > 0 and a negative sign if the LFC was < 0.

### Single microglia sequencing

Following microdissection, the hippocampi of 2 mice were pooled and placed in ice-cold FACS buffer (PBS, 2 mM EDTA and 2% fetal calf serum) with 5 μM actinomycin D to stop transcriptional activation due to the isolation procedure (Sigma-Aldrich, #A1410, St Louis, MO, USA). Subsequently, the tissue was mechanically triturated and enzymatically dissociated using the Neural Tissue Dissociation Kit (P; Miltenyi Biotec, Leiden, the Netherlands) following manufacturer's instructions, while adding 5uM actinomycin D to the samples. Samples were filtered through a 70 μm mesh strainer (BD2 Falcon) with FACS buffer (without actinomycin D) and centrifuged for 15 min at 500 *g* at 4°C. From this point forward, actinomycin D was omitted from any buffer to avoid toxicity. Then, cells were resuspended in 30% Percoll (GE Healthcare) and centrifuged for 15 min at 500 *g* at 4°C. After myelin and supernatant removal, the pelleted cells were resuspended in FcR blocking solution (Miltenyi) in cold FACS buffer, following manufacturer's instructions. After washing, cells were centrifuged at 500 *g* for 10 min at 4°C and incubated with the primary antibodies, BV421 rat anti-mouse CD11b (clone M1/70, #562605, BD Bioscience, 1/500) and APC rat anti-mouse CD45 (clone 30-F11, #559864, BD Bioscience, 1/500) and adding e780 (#65-0865-14, Thermo Fisher, 1/2,000) as a viability marker, for 40 min at 4°C. Moreover, unstained and single-stained matched control samples were used to assess autofluorescence and/or non-specific binding of the antibodies. Samples were run on the BD FACS Aria III Flow Cytometer or the BD Influx cell sorter operated by the KU Leuven FACS Core. Due to a low recovery of microglia from the 11M TAUtg and TAUwt samples, isolated mouse microglia were mixed with primary rat astrocytes to ensure high enough cell density for 10X Genomics single cell processing. Given the low yield, it was decided to run another *n* = 1 for the TAUtg and TAUwt samples, which again gave a low recovery. These mouse microglia were subsequently mixed with human H9 embryonic stem cell-derived monocytes to ensure the needed cell density for 10× Genomics.

Next, isolated microglia were prepared for single cell sequencing using the 10X Genomics single cell gene expression profiling kit. cDNA libraries were prepared following manufacturer's instructions, and all samples were sequenced in a single run on the Illumina

HiSeq 4000 platform with the 10X Genomics recommended sequencing specifications. The raw BCL files were demultiplexed and aligned with Cellranger (v3.1.0) against a joint database with the mouse genome (mm10, GRCm38), the rat genome (Rnor_6.0), and the human genome (GRCh38). Cells from each experimental group were clustered using Scanpy (v1.4.4; Wolf et al, 2018). Cells were separated, as expected, in clusters per organism. For each cluster with a clear mouse origin, the cell ids were stored. For consistency, we performed this step also on samples with no rat or human cells added. Subsequently, all cells were remapped against a mouse database (mm10, GRCm38) and only cells identified as mouse cells in the previous step were kept for analysis.

Cells with < 750 reads or with more than 5% of reads mapping to mitochondrial genes were filtered out. Cells with more than 30,000 genes were considered doublets and removed. Genes detected in < 100 cells were discarded. Per experimental group, we obtained the following number of cells: APPwt-4M: 700; APPwt-11M: 1,799; APPtg-4M: 991; APPtg-11M: 2,275; TAUwt-4M: 3,856; TAUwt-11M: 761; TAUtg-4M: 4,224; and TAUtg-11M: 1,503. The mean sequencing depth was 6,580 reads/cell, and on average, 2,335 genes were detected per cell.

Cells passing QC were analyzed using the Seurat (v.3.1.1) R package. The combined object (including all APP and TAU experimental groups) was split into a list, where every element is a batch (as referred to sorting date, corresponding to the 4 age*genotype groups +1 additional TAU-11M sorting run). Standard pre-processing (log normalization) was performed individually for each of the five batches, and variable features (nfeatures = 2,000) were identified based on a variance stabilizing transformation (selection.method = "vst"). Next, we identified anchors using the "FindIntegrationAnchors" function, giving the list of Seurat objects as input. We used all default parameters, including the dimensionality of the dataset (dims = 1:30). We passed these anchors to the "IntegrateData" function, to get an integrated (or "batch-corrected") expression matrix for all cells, enabling them to be jointly analyzed. Scaling was performed on all genes from the filtered, unintegrated matrix. We used the integrated matrix for downstream analysis and visualization using the standard Seurat workflow.

Unbiased pseudotime inference of microglia progression from HM to ARM/DAM was performed using the Scorpius (v1.0.4.1) R package using default parameters.

### Differential expression analysis (single microglia)

We performed differential expression analysis between activated response microglia (cluster ARM) and homeostatic microglia cluster 1 (HM.1) for each transgenic model, using Seurat's "FindMarkers" function with a Wilcoxon rank sum test, with Bonferroni correction for multiple testing. We used the unintegrated, filtered count matrix. Log fold changes > |0.2| with $P_{adj} < 0.05$ are considered significant. Pearson's correlation was calculated on log fold changes.

### Gene sets for determination of microglial cell state

We extracted publicly available marker sets of different microglial cell states: activated response microglia, cycling and proliferating microglia, homeostatic microglia, interferon-response microglia, and transitioning response microglia from Sala Frigerio et al (2019);

cytokine response microglia from Mancuso et al (2019a); and neutrophil–monocytes and macrophages from Friedman et al (2018); and a list of MHC markers can be found in Dataset EV6. For each microglial cell, we calculated the mean abundance levels of each gene in a marker set against the aggregated abundance of random control gene sets, using Seurat's "AddModuleScore" function. These data were visualized on UMAP embeddings to determine cellular states within the single cell clustering.

### Cell-specific datasets

Cell type-specific genes sets for pyramidal neurons (n = 701 genes), interneurons (n = 364), astrocytes (n = 239), microglia (n = 435), oligodendrocytes (n = 452), endothelial (n = 352), and ependymal cells (n = 483) were derived from Zeisel et al (2015) and for homeostatic and activated response microglia from Sala Frigerio et al (2019). Using these gene sets and a count matrix that was z-score normalized across samples, we calculated for each cell type (t) the average z-score ($Z_{tg}$) for each experimental group (g) and compared this to the respective 4M WT group. We assessed significance using empirically derived P-values (Appendix Fig S6).

### GWAS gene set enrichment analysis

Human GWAS genes derived from Marioni et al (2018) were converted to mouse orthologues using the Ensemble Biomart Release 94 (Zerbino et al, 2018). Using the gene set enrichment analysis preranked module (Broad Institute (Subramanian et al, 2005; Mootha et al, 2003), two-sided testing), enrichment for GWAS genes was tested among all genes, sorted from up- to downregulated genes based on signed log10(P-value) based on the different statistical comparisons (age, genotype, and age*genotype interaction), for APPtg and TAUtg separately, and performing BY P-value adjustment. Thus, GWAS gene enrichment is assessed based on gene rank and therefore impervious to varying numbers of significantly differentially expressed genes.

### Weighted gene co-expression network analysis (WGCNA)

The WGCNA package in R (Langfelder & Horvath, 2008) was used to build unsigned mRNA co-expression networks for APPtg and TAUtg mice separately using all 15824 expressed genes. To generate an adjacency matrix with is the smallest threshold that satisfies the scale-free topology fit at $R^2 = 0.9$, soft power 3 is used for APPtg and 4 for TAUtg mice, respectively. The topology overlap (TO) was calculated based on the adjacency matrix which measures the network interconnectedness. The topology overlap dissimilarity was then calculated by 1-TO and used as input for average linkage hierarchical clustering. Branches of the hierarchical clustering tree were then assigned into modules using cutreeHybrid from the dynamicTreeCut package (deepSplit = 2, minModuleSize = 30, Langfelder et al, 2008). The resulting 31 APPtg and 32 TAUtg modules were each summarized by the first principal component, known as module eigengenes (MEs). Next, Fisher's exact test with BY P-value adjustment was used to determine whether a list of cell type-specific genes overlap significantly with genes in a module. More than half of all modules (APPtg: 16/31; TAUtg: 20/32) show significant overlap with a specific cell type ($P_{adj} < 0.05$), particularly with the

neuron and interneuron gene sets (APPtg: 10/31; TAUtg: 11/32; see Appendix Figs S2 and S3). Gene set enrichment analysis of GWAS genes from Marioni et al (2018) at different P-value cut-offs among the WGCNA-derived modules in APPtg or TAUtg mice was performed using Fisher's exact test with BY P-value adjustment.

Functional annotations of the modules were performed using first GOrilla (Eden et al, 2009) and when no significant enrichment could be found, using DAVID (see Datasets EV2 and EV3 (Huang et al, 2009b,a)). GO categories were deemed significant if the FDR-corrected P-value (GOrilla) or Benjamini-based $P < 0.05$ (DAVID).

To search for potential regulators in each module, we ran i-cisTarget (Imrichová et al, 2015) which predicts transcription factor motifs (position weight matrices) and experimental data tracks (e.g., ENCODE, Roadmap Epigenomics Project) that are enriched in the input set of regions (i.e., genomic regions for each gene within the module). The default setting collected over 23,588 features across all the databases available in i-cisTarget. Only regulators that were also expressed within the same WGCNA module were considered. Top regulators were selected based on the maximum normalized enrichment score for feature enrichment.

**Gene set overlap assessment**

Overlap between the GWAS $P < 0.001$ gene set ($n = 314$), all genes significantly differentially expressed within the APPtg age*genotype interaction comparison ($n = 798$), genes within the APPtg-Blue module ($n = 4,236$), and the microglia-specific gene set ($n = 435$) were assessed using SuperExactTest (Wang et al, 2015; version 1.0.4), which calculates, based on combinatorial theory, the statistical probability of finding an over-representation of genes within the intersection of multiple sets, compared to random expectation. P-values were BY-adjusted (see Appendix Table S3).

**Empirical P-values associated with predicted changes in cell type fraction**

We predict changes in cell type fractions based on shifts in the mean (per gene z-normalized across samples) expression of a wide panel of genes associated with a specific cell type by Zeisel et al (2015), see Fig EV1). Given that we are interested in shifts of cell types when compared to the 4M wild-type (WT-4M) mice (from the corresponding strain), we use the mean gene expression in only the WT-4M mice as the baseline in the gene expression z-normalization. So, the z-normalized expression $Z_i$ of a gene $i$ in sample $s$ can be calculated from the (limma-voom normalized) expression $E_i$ as follows:

$$Z_{is} = \frac{E_{is} - \text{mean}(E_{i(WT\text{-}4M)})}{\text{std}(E_i)}$$

Then, the average z-score ($Z_{tg}$) for a cell type ($t$), in an experimental group ($g$) across all genes of a cell type gene set is calculated as the mean of the per gene z scores ($Z_{is}$) in the experimental group. It is the $Z_{tg}$ scores that are depicted in Fig EV1.

Next, an empirical P-value is assigned to each $Z_{tg}$ as follows: per cell type-specific gene set, 10,000 random gene sets (of the same size as the cell type-specific gene set) were sampled. For each random iteration, a $Z_{rtg}$ is calculated using the z-normalized $Z_{is}$

**The paper explained**

**Problem**
Over 40 genetic risk loci for Alzheimer's disease (AD) currently meet the genome-wide significance criterion, but recent evidence shows that hundreds of additional genetic variants that fall below this stringent cut-off still significantly contribute to the risk of AD. We wondered under which conditions these genes are expressed and in particular whether they respond to TAU or amyloid-β (Aβ) pathology.

**Results**
We have used bulk RNA-seq from 2 mouse models of AD, one displaying Aβ and the other TAU pathology, to assess which genes associated with GWAS-derived risk variants would change their transcriptional expression in function of genotype and age. We identified that many genes linked to AD risk in different GWAS studies are particularly responsive to Aβ but not to TAU pathology, and confirm with high statistical significance 11 novel risk genes below the genome-wide significance threshold that significantly upregulate their expression when facing increasing Aβ levels. All these genes are expressed in microglia. By using single microglia sequencing, we could confirm that microglia exposed to Aβ drastically switch their phenotype to an activated status, while this is seen to a much lesser extent in the TAU-expressing mice.

**Impact**
We conclude that the genetic risk of AD largely dictates the microglia response to Aβ pathology, placing genetic risk of AD downstream of amyloid pathology and upstream of TAU pathology.

scores in exactly the same way as described above. Of each population of randomized $Z_{rtg}$ values (per group $g$ and cell type $t$), normality is confirmed and a population mean and standard deviation are calculated to assign an empirical P-value to the observed $Z_{tg}$ using a cumulative distribution function (Appendix Fig S6). The combined observations across 6 experimental groups (excluding both WT-4M) and 6 different cell gene sets types are Bonferroni-corrected to exclude false-positive observations.

**Prediction of microglial contribution to the bulk RNA-seq data of the top 18 genes**

For each of the top 18 genes, the average expression level in all microglial cells was calculated, as well as their expression in HM (combining clusters HM.1 and HM.2) and all non-homeostatic microglia (termed "ARM$^+$" cells, consisting of all microglial clusters that were not HM.1 and HM.2), per experimental group (see Fig EV3-i for examples). Subsequently, the extend of overall microgliosis was estimated using the microglia marker genes as defined by Zeisel et al (2015), by calculating the average z-score normalized expression of all 436 genes in the bulk expression data, per experimental group and normalizing these values to the group with the lowest expression (APPwt_4M; see Fig EV3-ii). Multiplying the overall microgliosis by the fraction of HM and ARM$^+$ cells per experimental group (calculated from the single cell data, see Figs 6C and EV3-ii), an estimation was made of the overall contribution of genes expressing in HM and ARM$^+$ cells, including microgliosis per experimental group (see Fig EV3-iii). The predicted expression was then fitted against the observed

expression for that gene in the bulk RNA-seq data, and the regression coefficient $r^2$ and BY-adjusted $P$-value of the fit were used as an indication of how much of the observed bulk expression can be explained based on microglial expresssion (see Figs 6F and EV3-iv and -v).

### General statistical methods

In all instances of multiple testing, BY $P$-value adjustment was performed in Python (http://www.statsmodels.org/0.8.0/generated/statsmodels.sandbox.stats.multicomp.multipletests.html) and $\alpha = 0.05$ throughout the study. When using Fisher's exact test, one-sided $P$-value testing was performed; two-sided testing was performed in all other cases. More details for each individual statistical analysis can be found above.

## Data and software availability

All RNA-seq data have been submitted to the GEO database under accession number GSE110741 and GSE142267; other data are available in the Expanded View Datasets or the Appendix Tables. The software tools used for this study include Tophat2 (version 2.0.7, Trapnell *et al*, 2009), available from https://ccb.jhu.edu/software/tophat/index.shtml; Subread/Featurecounts (Liao *et al*, 2014), available from http://subread.sourceforge.net/; Pandas Python Data Analysis Library, available from http://pandas.pydata.org/; Limma/Linear Models for Microarray Data (Ritchie *et al*, 2015), available from https://bioconductor.org/packages/release/bioc/html/limma.html; Gene Set Enrichment Analysis (Mootha *et al*, 2003; Subramanian *et al*, 2005), available from http://software.broadinstitute.org/gsea/index.jsp; WGCNA package in R (Langfelder & Horvath, 2008), available from https://cran.r-project.org/web/packages/WGCNA/index.html; dynamicTreeCut package (Langfelder *et al*, 2008), available from https://cran.r-project.org/web/packages/dynamicTreeCut/index.html; Gene Ontology enrichment with GOrilla (Eden *et al*, 2009), available from http://cbl-gorilla.cs.technion.ac.il/; DAVID (Huang *et al*, 2009a,b), available from https://david.ncifcrf.gov/home.jsp; i-cisTarget (Imrichová *et al*, 2015), available from https://gbiomed.kuleuven.be/apps/lcb/i-cisTarget/; SuperExactTest (Wang *et al*, 2015), available from https://cran.r-project.org/web/packages/SuperExactTest/index.html; StatsModels for Python available from http://www.statsmodels.org/0.8.0/generated/statsmodels.sandbox.stats.multicomp.multipletests.html; Seurat available from https://cran.r-project.org/web/packages/Seurat/index.html (Stuart *et al*, 2019); and Scorpius available from https://cran.rstudio.com/web/packages/SCORPIUS/index.html (preprint: Cannoodt *et al*, 2016).

**Expanded View** for this article is available online.

### Acknowledgements

We thank Veronique Hendrickx and Jonas Verwaeren for animal husbandry and Carlo Sala Frigerio and Tom Jaspers for discussions. We thank in particular Prof. Mathias Jucker for making the line APP/PS1 available for our research and for providing us with an additional batch of aged APP/PS1 mice on very short notice. Confocal microscopy equipment was acquired through a Hercules Type 1 AKUL/09/037 to Wim Annaert. Annerieke Sierksma and Bart De Strooper are supported by the Opening the Future campaign of the Leuven Universitair Fonds (LUF) and the Alzheimer Research Foundation (SAO-FRA; P16017). Renzo Mancuso is a recipient of a postdoctoral fellowship from the Alzheimer's Association USA. Nicola Fattorelli is an Fonds voor Wetenschappelijk Onderzoek (FWO) fellow (1139520N). Evgenia Salta is an FWO fellow (12A5316N) and supported by the Alzheimer's Association (AARF-16-442853). Work in the De Strooper laboratory was supported by the FWO (grant no. G0C9219N), KU Leuven, VIB, and a Methusalem grant from KU Leuven and the Flemish Government, the "Geneeskundige Stichting Koningin Elisabeth", the MRC, the Alzheimer Society, and Alzheimer Research UK. Bart De Strooper is holder of the Bax-Vanluffelen Chair for Alzheimer's Disease. Luc Buée is supported by grant ANR-16-COEN-0007.

### Author contributions

AS, BDS, and MF designed the study and wrote the manuscript. AS, AL, JZ, NF, NT, and MF performed analyses. DB and LB provided TAUtg mice; ES, RM, DB, and LB provided expertise and feedback. All authors read and approved the final manuscript for publication.

### Conflict of interest

B.D.S. is *ad hoc* consultant for various companies but has no direct financial interest in the current study. L.B. is consultant for Servier and Remynd. He receives research funding from UCB Pharma, but not for the work presented in the current manuscript. A.S., A.L., E.S., R.M., J.Z., D.B., and M.F. report no biomedical financial interests or potential conflicts of interest.

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
