## [Review Process File · EMBO Molecular Medicine]

Novel Alzheimer risk genes determine the microglia response to amyloid- β but not to TAU pathology

Annerieke Sierksma, Ashley Lu, Renzo Mancuso, Nicola Fattorelli, Nicola Thrupp, Evgenia Salta, Jesus Zoco, David Blum, Luc Buée, Bart De Strooper, Mark Fiers

Review timeline:

Submission date:	15 March 2019
Editorial Decision:	30 April 2019
Revision received:	28 November 2019
Editorial Decision:	12 December 2019
Revision received:	20 December 2019
Accepted:	20 December 2019

Editor: Céline Carret

Transaction Report:

1st Editorial Decision

30 April 2019

Thank you for the submission of your manuscript to EMBO Molecular Medicine and I am so very sorry that it has taken so long to get the 2 reports we ask for. Nevertheless, we have now heard back from the referees whom we asked to evaluate your manuscript.

You will see that they find the data of interest and referee #2 find the work laudable! Still, referees have three main concerns that we would encourage you to address:

- the first and most important concern, shared between the two referees, concerns the cell population and bulk sequencing issue (1st comment of both referees). We believe that to fully elucidate whether changes come from gene expression or cell population composition, single cell sequencing should be performed. This option was also preferred by referee #2 during cross-commenting.
- 2nd point of ref. #1 regards APP as a model of sAD vs. mendelian AD, this point should be thoroughly discussed
- medical impact (ref. #2), validation of the findings in human tissue would increase the clinical significance of the study.

We would therefore welcome the submission of a revised version within three months for further consideration and would like to encourage you to address all the criticisms raised as suggested to improve conclusiveness and clarity. Please note that EMBO Molecular Medicine strongly supports a single round of revision and that, as acceptance or rejection of the manuscript will depend on another round of review, your responses should be as complete as possible.

Please also contact us as soon as possible if similar work is published elsewhere. If other work is published, we may not be able to extend the revision period beyond three months.

I look forward to receiving your revised manuscript.

***** Reviewer's comments *****

Referee #1 (Remarks for Author):

The goal of this study is to analyze if the transcriptome changes happening due to the APP and/or MAPT transgenic gene overlap with the known genetic architecture of AD risk. The author found that in the APP model there is an overexpression of AD risk genes. A lot of these genes are involved in immune response and inflammation and expressed in microglia. On the other hand, they did not see this effect on the MAPT (Tau) transgenic models. As a result, the authors conclude that the APP mouse can be an informative model to study AD, especially early pathological stages.

However, there are two major problems in this study. The first one is that, as the author indicates a lot of the upregulated genes are microglia genes. It is known that AD pathology, especially in mouse, is characterized by microgliosis. Therefore, all the results presented in this study could be driven by just change in brain cell population and not really by changes in gene expression. The authors have a couple of sections about cell population quantification and changes, but it is not clear how it fits in the current manuscript and whether they corrected for cell population in their analyses. The authors should use some of the already published and validated digital deconvolution methods to quantify the cell proportion in each sample and correct for that. Even better, would be that they perform some single cell RNA-seq experiments to confirm that the reported findings are really change in gene expression and not cell population proportions.

The second problem is that one of the main findings of this study is that the APP model is informative for sporadic AD. However, the APP is modeled based on the Mendelian AD genes. It is known that APP expression and/or activity does not really change in sporadic AD, and the Mendelian AD cases are not enriched for the risk factor described for sporadic AD (Single variants and PRS). Based on this, how do the authors explain their findings? A more in-depth discussion of this is needed.

Referee #2 (Comments on Novelty/Model System for Author):

Technical quality: high

The novelty of the presented data is medium. Microglia and neuroinflammation in the context of AD is an established phenomenon. Here the separation amyloid/tau is novel. Bulk tissue sequencing complicates interpretation of the data.

The medical impact is low, none of the genes identified are validated in mouse or human CNS tissue.

Referee #2 (Remarks for Author):

Sierksma et al. report on novel AD risk genes identified in AD-GWAS studies in amyloid and tau mouse models to delineate their involvement in AD. The attempt to bridge the gap between risk loci in GWAS data and real biology, as is presented here, is laudable.

There are a few conceptual and technical issues with the manuscript that require additional clarification in order to substantiate the observations and claims made.

1) It is unclear, why in this day and age, the authors only performed hippocampal tissue RNAseq. Sorting/isolation protocols for different CNS cell types, and especially microglia are established. In addition, microglia display heterogeneity in AD, as has been reported by several labs (i.e. Keren-Shaul, Cell, 2017).

2) Bulk tissue RNA-seq data is hampered by changes in cellular composition of the tissue, microglia numbers possibly change during AD etiology, potentially skewing expression data, were the transcriptomes myeloid balanced?

- 3) None of the deregulated genes is validated in CNS tissue of these mice by (F)ISH, immunostaining, or even better in human AD tissues.
- 4) What is the added value of these presented data in relation to the seminal Zhang Cell paper and subsequent papers elucidating the involvement of microglia in AD pathology, that is not clear.
- 5) What is the expression behaviour of the 18 "core" genes in other AD data sets? Like in the Keren-Shaul data and in other (total tissue) AD expression profiles.
- 6) Of these 18 genes, and other key genes, dot plots with their expression values in the different samples should be provided, potentially as supplemental data.
- 7) In fig2, LFC are plotted, but the direction of the changes are not provided, in $2A < I$ assume a positive LFC means higher expression in aged animals?
- 8) In Fig 2, for example panel B, it is unclear why are some of the genes are differential in APP only (like Ccl3) and a gene with quite similar LFCs is differential in both (Olf111) APP and Tau.
- 9) The WGCNA analysis was performed using an unsigned network. Horvath and Langfelder in their paper state that signed networks are generally (90% of the cases) superior when analysing biological data. Why was an unsigned network used, as the direction of gene expression changes is arguably relevant in these data.
- 10) The blue model is enriched for microglia genes (Fig 4A). The most enriched txn factor in these genes is Spi1/Pu.1, Fig 4C. This is not very surprising as Spi1 is a key txn factor for microglia, so additional clarification as to why this is relevant for AD would help, in addition to a protective snp in the SPI1 gene.
- 11) What is the difference between "general" and "homeostatic" gene sets, line 271
- 12: What was used as input for icisTarget? As the output leads to Fig 4, was it the genes of the blue module?

1st Revision - authors' response

28 November 2019

We would like to thank the reviewers for their constructive feedback and criticism. We have adapted and extended the manuscript according to their suggestions with changes to the previous version highlighted in grey. A detailed response to the reviewer comments can be found below.

Referee #1 (Remarks for Author):

The goal of this study is to analyze if the transcriptome changes happening due to the APP and/or MAPT transgenic gene overlap with the known genetic architecture of AD risk. The author found that in the APP model there is an overexpression of AD risk genes. A lot of these genes are involved in immune response and inflammation and expressed in microglia. On the other hand, they did not see this effect on the MAPT (Tau) transgenic models. As a result, the authors conclude that the APP mouse can be an informative model to study AD, especially early pathological stages.

*However, there are **two major problems** in this study. The first one is that, as the author indicate a lot of the upregulated genes are microglia genes. It is known that AD pathology, especially in mouse, is characterized by microgliosis. Therefore, all the results presented in this study **could be driven by just change in brain cell population and not really by changes in gene expression**. The authors have a couple of sections about cell population quantification and changes, but it is not clear how it fits in the current manuscript and whether they corrected for cell population in their analyses. The authors should use some of the already published and validated digital deconvolution methods **to quantify the cell proportion in each sample and correct for that**. Even better, would be that they **perform some single cell RNA-seq** experiments to confirm that the reported findings are really change in gene expression and not cell population proportions.*

We agree with this comment. We have now added a single cell transcriptomics microglia experiment of our APP/PS1 and TAU22 mouse models, which will be submitted to the GEO database very shortly. We have integrated the single microglia RNA-seq with our bulk analysis showing indeed a mixed picture. The observed expression changes in 16 of the 18 top genes can be well explained from the combination of changes in gene expression and numbers of microglia. The results from this experiment can be found in the manuscript in Fig.6, in Suppl. Fig.S5-7 and are described from page 12 onwards in the results section. To summarize, we demonstrate that the distribution of microglia subtypes over different cell states is dramatically different between the APPtg mice and the TAUtg mice (Fig.5D). The results confirm that the A β pathology drives the activated response microglia (ARM; similar but not identical to DAM) response already very early in the pathogenesis (from 4M onwards) even before any behavioral or neurodegeneration is observed. The Tau mice show much less activated microglia and, importantly, only at 11M. We surmise that they appear secondary to neuronal damage caused by TAU. We make this conclusion based on the expression analysis in Suppl. Fig.S2 which demonstrates that early and late changes in TAUtg mice are mostly in neuronal and synaptic genes, while in the APPtg the immune response is early, and is amplified hugely over time. Thus, the single cell data confirm our previous bulk sequencing conclusions and confirm that the microglia response is not only quantitative but also qualitative in nature.

We refer to the two new paragraphs at the end of the result sections which cover the new findings for further discussion and analysis of the findings. We feel that the manuscript significantly improved by this suggestion and thank the referee for this critical input.

As a further note for the referee we have tried to estimate the degree of microgliosis per experimental group, trying two published tools, Cibersort⁵ and MuSiC⁶, to deconvolute the proportions of the different cell types (using Zeisel et al.⁷ as a reference for the different cell types), however, this proved to be unsuccessful. In fact, according to Cibersort, none of the experimental groups contained microglia, while MuSiC only detected microglia in the APPtg-10M group (see Reviewer Fig.1 below). Thus, these tools were, in our experimental setting, not suitable to make reliable estimations of the proportion of microglia in the brain from bulk RNA-seq data. We have therefore employed our own method to estimate fractional changes in the number of microglia within each experimental group. A full description can be found in the Materials & Methods section (p.27) and examples of our calculations for three genes (*ApoE*, *Inpp5d* and *Chu*) can be found in Supplementary Fig.7. A summary of the results for each of the top18 genes can be found in Fig.6F, demonstrating that the microglial expression of *Chu* and *Slc2a5* cannot explain the observed change in the bulk RNA-seq data, while the microglial contribution to the bulk RNA-seq data for the other 16 genes is significant and substantial (>67% explained variation, see Fig.6F).

The main difference between our approach and that of Cibersort and MuSiC is that the latter two methods attempt to estimate percentages of all cell types, whereas we only predict fractional changes of the microglia. Given that we use our fractional changes in a regression model, we do not need absolute percentages. Moreover, the single cell data used to make our predictions for the bulk RNAseq is derived from the same experimental groups, making the predictions more trustworthy.

Reviewer Fig.1: outcome of the online Cibersort tool (left) and the MuSiC algorithm (right), using as input all 94 samples across 8 experimental groups and the marker genes for the different cell types listed, as defined by Zeisel et al.⁷ As can be seen, for most experimental groups 0% of microglia are detected according to both softwares.

The second problem is that one of main findings of this study is that the APP model is informative for sporadic AD. However, the APP is modeled based on the Mendelian AD genes. It is known that APP expression and/or activity does not really change in sporadic AD, and the Mendelian AD cases are not enriched for the risk factor described for sporadic AD (Single variants and PRS). Based on this, how the authors explain their findings? A more depth discussion of this is needed.

We apologize for not having explained this better. We used the FAD mouse models here to model key neuropathological hallmarks of AD, i.e. amyloidosis and neurofibrillary tangles and we ask how the brain responds to these challenges. Amyloid plaques in FAD are neuropathologically similar to those in sporadic AD (although their different ages of onset may affect the level of severity^{8,9}). We see no reason why the amyloid pathology in FAD and SAD would induce a different response, while we agree that the upstream causes of the accumulation of A β might be different. Here, we use an amyloidosis model to study the nervous system's cellular and transcriptomic response to this biochemical insult and compare it with a TAU/tangle insult. The TAUtg is indeed not strictly a model for FAD but for FTD (which we acknowledge in the paper). Also in that model we use the TAU biochemical pathology to study the gene expression responses again assuming that the cellular responses to Tau tangles will be to a certain extent generic in nature. The null hypothesis was that both A β and TAU biochemical changes would induce similar gene expression profile changes in glia cells and neurons, because previous work, in which our group was involved, has shown that the two models show very similar cognitive alterations¹⁰. Interestingly, our data show that the molecular pathology caused by the two different biochemical lesions cause very different gene expression alterations throughout the brain. This is interesting as it helps us to start to dissect further the cellular phase of disease and shows that the same cognitive symptoms can come from very different pathological mechanisms.

We believe that the cellular responses in sporadic disease on progressive A β or TAU accumulation will be very similar to the cellular responses in FAD as the driving biochemical lesions are very similar. We have addressed this issue in the introduction (p4., lines 70-73).

Referee #2 (Comments on Novelty/Model System for Author):

Technical quality: high

The novelty of the presented data is medium. Microglia and neuroinflammation in the context of AD is an established phenomenon. Here the separation amyloid/tau is novel. Bulk tissue sequencing complicates interpretation of the data.

The medical impact is low, none of the genes identified are validated in mouse or human CNS tissue.

We accept the criticism of the referee, although we would like to point out that all of our genes were previously implicated in human GWAS studies and are part of the polygenic risk scores (PRS) as calculated by the Cardiff group^{11,12}. We used the two mouse models to see how these human candidate, and subthreshold candidate genes behave in these (in our opinion) highly relevant mouse models to the disease. Thus, our work is certainly not disconnected from the clinical observations and provides functional evidence that also sub-significant results coming from GWAS studies make functional significance. We are convinced that this approach, combining human genetics with mouse functional studies, brings a novel dimension to the field and shows a way forward in how to understand GWAS implicated genes in a functional relevant context. To his/her point of single cell resolution, we have now added a series of single cell sequencing data (all in figure 6 and Supplementary Fig.S5-S7, which is entirely new data).

Referee #2 (Remarks for Author):

Sierksma et al. report on novel AD risk genes identified in AD-GWAS studies in amyloid and tau mouse models to delineate their involvement in AD. The attempt to bridge the gap between risk loci in GWAS data and really biology, as is presented here, is a laudable.

We feel supported by this comment, which is in line with what we describe above.

There are a few conceptual and technical issues with the manuscript that require additional clarification in order to substantiate the observations and claims made.

1) It is unclear, why in this day and age, the authors only performed hippocampal tissue RNAseq. Sorting/isolation protocols for different CNS cell types, and especially microglia are established. In addition, microglia display heterogeneity in AD, as has been reported by several labs (i.e. Keren-Shaul, Cell, 2017).

This study was started a long time before the single cell data became available. We still believe that the information coming from this bulk sequencing story is highly informative. Nevertheless, we have now, as suggested by the reviewers, performed additional single microglia RNA sequencing of all 8 experimental groups to improve the novelty of our manuscript and to be able to assess the difference in response in APPtg and TAUtg mice with aging. We urge the reviewer to also read our response to the first comment of Reviewer #1.

2) Bulk tissue RNA-seq data is hampered by changes in cellular composition of the tissue, microglia numbers possibly change during AD ethology, potentially skewing expression data, were the transcriptomes myeloid balanced?

The different experimental groups indeed have very different loads of microglia, as we also discussed in the previous version of the manuscript and documented in Suppl. Fig.S2, with the APPtg-10M mice having the highest proportion of microglia. However, while this can alter the gene expression, it is part of the cellular response to the biochemical lesions, and it is very interesting to see how microgliosis becomes a major part of the response very early on in the APPtg model, but not in the TAUtg model.

We have now also performed single microglia RNA-seq to gain further insight into the different microglial responses in APPtg and TAUtg mice. We demonstrate that microglia in APPtg mice massively undergo a phenotypic shift from homeostatic microglia to ARM, while this effect is very mild in TAUtg mice. As can be seen in Fig.6E, the ARM cells are highly similar between APPtg and TAUtg mice, in agreement that the ARM (similar but not identical to DAM) response is a generic response to brain damage. It is, however, highly interesting that this phenotypic response is already initiated very early in the APPtg mice, suggesting that the microglia response is part of the initial response, turning A β accumulation to a cellular disease. In the TAUtg mice the microglia response comes late, when many other molecular responses (mainly in the neurons) have already been downregulated (see Suppl. Fig.S2). Thus, in TAUtg the microglia response is more downstream, likely a response on the neuronal damage and also much less pronounced, to the extent that in the bulk sequencing only very limited inflammation is contributing to the overall responses (Suppl.Fig.S2)

Using the single cell microglia RNA-seq data we have also made predictions of the microglial contribution to the observed bulk RNA-seq data. For a detailed description, we refer to our response to the first question of reviewer #1 and to the results section from p.13 and the Materials & Methods section on p.27.

3) None of the deregulated genes is validated in CNS tissue of these mice by (F)ISH, immunostaining, or even better in human AD tissues.

This question is largely addressed by the single cell sequencing data now added to the manuscript, and by Supplemental Fig.S6C showing the expression of the top 18 genes in different public datasets of various mouse models and human cases, both in bulk RNAseq and isolated microglia/single microglia sequencing. Our experiments and those publicly available ones demonstrate that indeed our top 18 genes are expressed in microglia (although some are lowly expressed, e.g. *Clu*, *Gpc2* and *Trem12*). We felt that again demonstrating that these genes are expressed by microglia in situ would not yield extra information at this stage.

4) What is the added value of these presented data in relation to the seminal Zhang Cell paper and subsequent papers elucidating the involvement of microglia in AD pathology, that is not clear.

We wish to emphasize that the point of this paper was not to identify that microglia and neuroinflammation are at the heart of transcriptomic changes in AD. With this paper we aimed to bridge the gap between genetic research on variants contributing to AD (polygenic) risk and the functional interpretation and basic understanding of how these variants modulate AD risk. The novelty in this study is the different involvement of AD risk variants in the transcriptomic response to amyloid and TAU pathology. The Zhang paper undoubtedly highlights the central role of the

immune system in the transcriptomic profile of late onset patients, but given that their patients had both pathologies, they could not distinguish whether the neuroinflammation was induced by amyloid or TAU pathology. Furthermore they could not deduce whether the responses are early or late, and finally their paper boiled down to a few genes that reached statistical relevance, while in our paper (logically, since we can fully control genetic background, tissue and age, in contrast to human studies) has yielded a plethora of novel interesting genes.

5) What is the expression behaviour of the 18 "core" genes in other AD data sets? Like in the Keren-Shaul data and in other (total tissue) AD expression profiles.

We have assessed the expression of our top 18 genes in our own microglia RNA-seq data, as well as in various other studies using bulk RNA-seq in amyloid mouse models and AD patients or who used isolated microglia for sequencing (including Keren-Shaul et al.¹ and Sala Frigerio et al.¹³). As can be viewed in Suppl. Fig.S6C, most of our top 18 genes are also upregulated in bulk RNA-seq data from a different amyloidosis mouse model (PS2APP) as well as in the cortex of AD patients. Sequencing of isolated microglia, however, confirms our observation that many of the top 18 genes remain unaltered or have slightly decreased expression in the single cells and are thus paradoxically down regulated in ARM microglia.

As described above, using regression analysis we have assessed to what extent the increased expression of a particular gene in our bulk-seq data can be explained by either changes in the number of microglia or the change in expression as a result of a cell state shift. As explained in more detail in the response to question 1 of Reviewer #1, we can demonstrate that for most genes, the increased expression in the bulk can be predominantly explained by both a shift in cell state as well as increased number of microglia, demonstrating that the global inflammatory response to A β is comprised of two microglial responses.

6) Of these 18 genes, and other key genes, dot plots with their expression values in the different samples should be provided, potentially as supplemental data.

We have added both the expression data for the top 18 genes both from the bulk RNAseq as well as from the single microglia RNA-seq data as Suppl. Fig.S6A+B.

7) In fig2, LFC are plotted, but the direction of the changes are not provided, in $2A < I$ assume a positive LFC means higher expression in aged animals?

Indeed, when a $\log_2(\text{fold change})/\text{LFC}$ is positive, it means the expression of the gene is higher in comparison to the baseline group (the 4M mice in case of the aging comparison in Fig.2A). Conversely, if the LFC is negative, the gene is expressed less than in the baseline group. We have clarified this in the figure legend of Fig.2A-C (p.43), adding sentences such as "Thus, genes with a positive LFC are more highly expressed in 10M mice over 4M mice" to aid the reader in the interpretation of those graphs.

8) In Fig 2, for example panel B, is it unclear why are some of the genes are differential in APP only (like Ccl3) and a gene with quite similar LFCs is differential in both (Olfr111) APP and Tau.

If the variability within the groups is very small, then very small LFCs may still be significantly different between the experimental groups. However, if the variability is larger, the difference in LFC between the 2 experimental groups that are being compared needs to be stronger, before they reach statistical significance. Biologically this is counterintuitive, but statistically it is necessary to respect the chosen test and not change the interpretation a posteriori.

9) The WGCNA analysis was performed using an unsigned network. Horvath and Langfelder in their paper state that signed networks are generally (90% of the cases) superior when analysing biological data. Why was an unsigned network used, as the direction of gene expression changes is arguably relevant in these data.

Actually Horvath and Langfelder themselves use both signed as well as unsigned networks alike to deconvolute patterns of coexpression in their transcriptomics data (e.g. see^{14,15} for examples of the unsigned network). We have reviewed 15 papers from 2008 onwards who used WGCNA for transcriptomic data analysis from mouse or human brain¹⁴⁻²⁷ and out of these only 3 specifically

specify they use the signed network^{16,22,26}. In fact, one of the most seminal transcriptomic papers in AD by Bin Zhang et al. in Cell 2013²⁷, who co-authored the initial paper on WGCNA²⁸, states that “The weighted network analysis begins with a matrix of the Pearson correlations between all gene pairs, ..”, thus suggesting an unsigned network was used. In our opinion, co-regulation of genes is not only uni-directional and the upregulation of one gene can simultaneously induce the downregulation of another. Given that both up- and downregulation of genes are necessary to ensure the activation/silencing of specific pathways, we decided to follow the trends in the field and use the unsigned network, to assess the co-regulation of up- and downregulated genes in totality.

10) *The blue model is enriched for microglia genes (Fig 4A). The most enriched txn factor in these genes is SPi1/Pu.1, Fig 4C. This is not very surprising as Spi1 is a key txn factor for microglia, so additional clarification as to why this is relevant for AD would help, in addition to a protective snp in the SPI1 gene.*

Indeed, it may not be surprising that Spi1 pops out as the key regulator of the APptg-Blue module, given that this module is enriched in microglia. We do wish to emphasize that the AD field lacks a good functional understanding of how PRS contributes to AD development. The observation in Fig 4C that Spi1 and other microglia-related transcription factors are predicted to be the top regulators of a large set of GWAS genes ($p < 0.001$, $n = 314$), highlights that the microglial enrichment of GWAS variants goes beyond the genome-wide variants, and is also observed among the polygenic risk. It highlights with a different method that our polygenic risk for AD is likely to affect how well our microglia react to AD-related biochemical insults, particularly amyloid. We have removed, however, the emphasis of Spi1 as a regulator of the APptg-Blue module taking into account the remark of the referee.

11) *What is the difference between "general" and "homeostatic" gene sets, line 271*

The general microglia set was based on the gene set as identified by Zeisel et al. 2016 ($n = 436$), which was aimed to find marker genes that would differentiate a microglia from another brain cell. The homeostatic microglia gene set was based on the gene set produced by Sala Frigerio et al. 2019 ($n = 396$), which describes a ‘resting’ or homeostatic cell state in contrast to an activated or ARM cell state. Thus, whereas the set from Zeisel et al. tries to differentiate the microglia from other cell types, the homeostatic gene set tries to differentiate different microglia states. The sets also only have 75 genes in common, including *Aif1*, *Cx3cr1*, *Inpp5d* and *Csf1r*.

Given that we now have single microglia RNAseq data to add to the paper, we have removed this section from the paper, as we feel these previous estimations are less informative than the newly added single microglia sequencing data.

12) *What was used as input for icisTarget? As the output leads to Fig 4, was it the genes of the blue module?*

As stated in the Materials & Methods section, line 642-645, the input for i-cisTarget were the genomic regions for each gene within each module. Thus, the enrichment for specific transcription factors was assessed on the level of each module, assessing all 63 modules in totality. Next, we narrowed down the regulatory transcription factors per module to those that were also contained within that module. The top regulators were selected based on the maximum normalized enrichment score. As can also be seen in Supplementary Fig.S3 and S4, if a particular module was enriched for specific transcription factors that were also contained within that module, they are listed in red.

References

- 1 Keren-Shaul H, Spinrad A, Weiner A, et al. A Unique Microglia Type Associated with Restricting Development of Alzheimer’s Disease. *Cell* 2017; **169**: 1276-1290.e17.
- 2 Krasemann S, Madore C, Cialic R, et al. The TREM2-APOE Pathway Drives the Transcriptional Phenotype of Dysfunctional Microglia in Neurodegenerative Diseases. *Immunity* 2017; **47**: 566-581.e9.
- 3 Friedman BA, Srinivasan K, Ayalon G, et al. Diverse brain myeloid expression profiles reveal distinct microglial activation states and aspects of Alzheimer’s disease not evident in mouse models. *Cell Rep* 2018.

- 4 Sala Frigerio C, Wolfs L, Fattorelli N, Perry VH, Fiers M, De Strooper B. The Major Risk Factors for Alzheimer's Disease: Age, Sex, and Genes Modulate the Microglia Response to A β Plaques. *CellReports* 2019; **27**: 1293-1306.e6.
- 5 Newman AM, Steen CB, Liu CL, *et al.* Determining cell type abundance and expression from bulk tissues with digital cytometry. *Nat Biotechnol* 2019. DOI:10.1038/s41587-019-0114-2.
- 6 Wang X, Park J, Susztak K, Zhang NR, Li M. Bulk tissue cell type deconvolution with multi-subject single-cell expression reference. *Nat Commun* 2019; **10**. DOI:10.1038/s41467-018-08023-x.
- 7 Zeisel A, Machado ABM, Codeluppi S, *et al.* Cell types in the mouse cortex and hippocampus revealed by single-cell RNA-seq. *Science (80-)* 2015; published online Feb 19. DOI:10.1126/science.aaa1934.
- 8 Lippa CF, Saunders AM, Smith TW, *et al.* Familial and sporadic Alzheimer's disease: Neuropathology cannot exclude a final common pathway. *Neurology* 1996; **46**: 406–12.
- 9 Nochlin D, Van Belle G, Bird D, Sumi SM. Comparison of the severity of neuropathologic changes in familial and sporadic Alzheimer's disease. *Alzheimer Dis Assoc Disord* 1993; **7**: 212–22.
- 10 Lo AC, Iscru E, Blum D, *et al.* Amyloid and tau neuropathology differentially affect prefrontal synaptic plasticity and cognitive performance in mouse models of Alzheimer's disease. *J Alzheimer's Dis* 2013; **37**: 109–25.
- 11 Escott-Price V, Myers AJ, Huentelman M, Hardy J. Polygenic risk score analysis of pathologically confirmed Alzheimer disease. *Ann Neurol* 2017; **82**: 311–4.
- 12 Escott-Price V, Sims R, Bannister C, *et al.* Common polygenic variation enhances risk prediction for Alzheimer's disease. *Brain* 2015; **138**: 3673–84.
- 13 Sala Frigerio C, Wolfs L, Fattorelli N, *et al.* The major risk factors for Alzheimer's disease: Age, Sex and Genes, modulate the microglia response to A β plaques. *Cell Rep* 2019; **In Press**.
- 14 Oldham MC, Konopka G, Iwamoto K, *et al.* Functional organization of the transcriptome in human brain. *Nat Neurosci* 2008; **11**: 1271–82.
- 15 Miller J a, Horvath S, Geschwind DH. Divergence of human and mouse brain transcriptome highlights Alzheimer disease pathways. *Proc Natl Acad Sci U S A* 2010; **107**: 12698–703.
- 16 Swarup V, Hinz FI, Rexach JE, *et al.* Identification of evolutionarily conserved gene networks mediating neurodegenerative dementia. *Nat Med* 2018; **25**: 152–64.
- 17 Salih DA, Bayram S, Guelfi S, *et al.* Genetic variability in response to amyloid beta deposition influences Alzheimer's disease risk. *Brain Commun* 2019; **1**. DOI:10.1093/braincomms/fcz022.
- 18 Jones L, Lambert J-C, Wang L-S, *et al.* Convergent genetic and expression data implicate immunity in Alzheimer's disease. *Alzheimer's Dement* 2015; **11**: 658–71.
- 19 Allen M, Wang X, Burgess JD, *et al.* Conserved brain myelination networks are altered in Alzheimer's and other neurodegenerative diseases. *Alzheimer's Dement* 2018; **14**: 352–66.
- 20 Holtman IR, Raj DD, Miller JA, *et al.* Induction of a common microglia gene expression signature by aging and neurodegenerative conditions: a co-expression meta-analysis. *Acta Neuropathol Commun* 2015; **3**. DOI:10.1186/s40478-015-0203-5.
- 21 Matarin M, Salih D a, Hardy J, *et al.* Resource A Genome-wide Gene-Expression Analysis and Database in Transgenic Mice during Development of Amyloid or Tau Pathology Resource A Genome-wide Gene-Expression Analysis and Database in Transgenic Mice during Development of Amyloid or Tau Pathology. *Cell Rep* 2015; **10**: 633–44.
- 22 Hawrylycz M, Miller JA, Menon V, *et al.* Canonical genetic signatures of the adult human brain. *Nat Neurosci* 2015; **18**: 1832–44.
- 23 Tasic B, Menon V, Nguyen TN, *et al.* Adult mouse cortical cell taxonomy revealed by single cell transcriptomics. *Nat Neurosci* 2016; **19**: 335–46.
- 24 Miller JA, Oldham MC, Geschwind DH. A Systems Level Analysis of Transcriptional Changes in Alzheimer's Disease and Normal Aging. *J Neurosci* 2008; **28**: 1410–20.
- 25 Wang M, Roussos P, McKenzie A, *et al.* Integrative network analysis of nineteen brain regions identifies molecular signatures and networks underlying selective regional vulnerability to Alzheimer's disease. *Genome Med* 2016; **8**. DOI:10.1186/s13073-016-0355-3.
- 26 Seyfried NT, Dammer EB, Swarup V, *et al.* A Multi-network Approach Identifies Protein-Specific Co-expression in Asymptomatic and Symptomatic Alzheimer's Disease. *Cell Syst* 2017; **4**: 60-72.e4.

- 27 Zhang B, Gaiteri C, Bodea L-GG, *et al.* Integrated systems approach identifies genetic nodes and networks in late-onset Alzheimer's disease. *Cell* 2013; **153**: 707–20.
- 28 Zhang B, Horvath S. A general framework for weighted gene co-expression network analysis. *Stat Appl Genet Mol Biol* 2005; **4**. DOI:10.2202/1544-6115.1128.

2nd Editorial Decision

12 December 2019

Thank you for the submission of your revised manuscript to EMBO Molecular Medicine. We have now received the enclosed report from the referee who was asked to re-assess it. As you will see the reviewer is now globally supportive and I am pleased to inform you that we will be able to accept your manuscript pending minor editorial amendments and the comments from referee #2.

Please submit your revised manuscript within two weeks.

I look forward to reading a new revised version of your manuscript as soon as possible.

***** Reviewer's comments *****

Referee #2 (Comments on Novelty/Model System for Author):

Technical quality: high

The novelty of the presented data has increased by carefully intersecting data with GWAS results and adding scSeq data to address tissue composition issues.

The medical impact is medium, these are amyloid and TAU mouse models, if the observed changes induced by amyloid and Tau translate to the human situation remains to be determined.

Referee #2 (Remarks for Author):

Sierksma and co-workers significantly improved their manuscript in response to the raised comments. A few minor comments remain:

- 1) in the text the referral to fig 6, line 245 should presumably be fig.5.
- 2) In the pseudotime analysis, what was used at the start node?
- 3) Was the pseudotime done on all cells from WT, APP and TAU mice combined? How do the trajectories look like when only the WT+APP or WT+TAU samples are analysed? Is there a trajectory in TAU samples for is it largely driven by amyloid?
- 4) ARM microglia seem like an endstage. Is there any evidence of a trajectory or stepwise development from HM.1-HM.2-ARM? Like the DAM1 and DAM2 stages proposed by Keren-Shaul? Are there genes that are differentially expressed along the trajectory and identify/label stages along the pseudotime?
- 5) a bar graph illustrating the relative contribution of the different conditions/samples to the 9 identified clusters would facilitate interpretation of the cluster contribution & composition.

2nd Revision - authors' response

20 December 2019

***** Reviewer's comments *****

Referee #2 (Comments on Novelty/Model System for Author):

Technical quality: high

The novelty of the presented data has increased by carefully intersecting data with GWAS results and adding scSeq data to address tissue composition issues.

The medical impact is medium, these are amyloid and TAU mouse models, if the observed changes induced by amyloid and Tau translate to the human situation remains to be determined.

We thank the referee for acknowledging our efforts to improve the manuscript by adding the single microglia sequencing data. We agree that a similar approach in human samples would be invaluable; this is something we are currently working on in the lab.

Referee #2 (Remarks for Author):

Sierksma and co-workers significantly improved their manuscript in response to the raised comments. A few minor comments remain:

1) in the text the referral to fig 6, line 245 should presumably be fig.5.

We apologize for this error and have corrected this in the final version of the manuscript.

2) In the pseudotime analysis, what was used at the start node?

We have used the Scorpius package (<https://www.biorxiv.org/content/biorxiv/early/2016/10/06/079509.full.pdf>) to perform the trajectory inference analysis, which does not require the definition of a start node (minimizing the possibility of introducing a bias). The predicted trajectory analysis from Scorpius actually chose activated microglia as the starting point and progressed towards the homeostatic microglia. For ease of interpretation we have inverted the trajectory manually (by taking 1-pseudotime).

3) Was the pseudotime done on all cells from WT, APP and TAU mice combined? How do the trajectories look like when only the WT+APP or WT+TAU samples are analysed? Is there a trajectory in TAU samples for is it largely driven by amyloid?

Indeed, the pseudotime analysis was performed on all cells from WT, APPTg and TAUtg mice combined. As suggested by the referee we have now also performed the trajectory inference analysis separately on the APPTg & APPwt as well as on the TAUtg & TAUwt groups (see Reviewer Fig.1 below). For both the APPTg and TAUtg mice independently, a trajectory from HM to ARM cells (after, if required, pseudotime inversion) was predicted. This trajectory is clearly much more outspoken in the APPTg mice, probably because the abundance of ARM cells is much higher. For this visualization in Reviewer Figure 1 we have taken the most differentially expressed genes between HM and ARM cells in APPTg and TAUtg mice (as described on p. 13, lines 307-314 and in Fig. 6E). We selected the top 20 genes most differentially expressed in ARM and in HM cells, taking genes overlapping between the two models. In our opinion this analysis demonstrates that both mouse models show a similar transcriptomic transition from HM to ARM cells, although the results from the combined trajectory analysis may be driven somewhat more strongly by the APPTg mice.

4) ARM microglia seem like an endstage. Is there any evidence of a trajectory or stepwise development from HM.1-HM.2-ARM? Like the DAM1 and DAM2 stages proposed by Keren-Shaul? Are there genes that are differentially expressed along the trajectory and identify/label stages along the pseudotime?

When viewing Fig.6D it can be observed that the proportion of HM.2 cells increases as the pseudotime trajectory moves more towards the ARM side. This can be viewed in all experimental groups, thus we can indeed conclude that a transition occurs from HM.1 to HM.2, and then further towards ARM. We have not performed a differential expression on the pseudotime, but performed differential expression on the identified microglia subtypes (as HM.1, HM.2 and ARM). Upon analysis of the DE between HM.1 and HM.2 it was clear that genes upregulated in HM.2 overlap with genes upregulated in ARMs when compared to HM.1 + HM.2 (data not shown), again suggesting that HM.2 is an intermediate state between HM.1 and full activation. A more in depth analysis on a similar pseudotime trajectory can be found in Sala Frigerio et al. 2019 Cell Reports 27, p.1293-1306.

5) a bar graph illustrating the relative contribution of the different conditions/samples to the 9 identified clusters would facilitate interpretation of the cluster contribution & composition.

We thank the referee for this suggestion and have added this bar graph to the Appendix Figures (Appendix Fig.S5). This figure again highlights that the main experimental group contributing to the ARM cells is the APptg-11M group (43% of ARM cells).

Corresponding Author Name: Bart De Strooper & Mark Fiers

Journal Submitted to: EMBO Mol Med

Manuscript Number: EMM-2019-10606-V2